# The impact of climate oscillations on the surface energy budget over the Greenland Ice Sheet in a changing climate

Tiago Silva[1], Jakob Abermann[1,3], Brice Noël[2], Sonika Shahi[1], Willem Jan van de Berg[2], and Wolfgang Schöner[1,3]

[1]Institute of Geography and Regional Science, Graz University, Austria
[2]Institute for Marine and Atmospheric Research, Utrecht University, Netherlands
[3]Austrian Polar Research Institute, Vienna, Austria

**Correspondence:** Tiago Silva (tiago.ferreira-da-silva@uni-graz.at)

**Abstract.** Climate change is particularly strong in Greenland primarily as a result of changes in transport of heat and moisture from lower latitudes. The atmospheric structures involved influence the surface mass balance (SMB) of the Greenland Ice Sheet (GrIS) and their patterns are largely explained by climate oscillations which describe the internal climate variability. By using k-means clustering, we name the combination of the Greenland Blocking Index and the North Atlantic Oscillation index with the vertically integrated water vapor as NAG. NAG captures the influence of atmospheric circulation patterns from the North Atlantic on Greenland with the optimal solution of three clusters (positive, neutral and negative phase). With the support of a polar-adapted regional climate model, typical climate features marked under certain NAG phases are inter-seasonally and regionally analyzed in order to assess the impact of large-scale systems from the North Atlantic on the surface energy budget (SEB) components over the GrIS.

Given the pronounced summer mass loss in recent decades (1991–2020), we investigate spatio-temporal changes on SEB components within NAG phases in comparison to the reference period 1959–1990. We report significant atmospheric warming and moistening across all NAG phases. The pronounced atmospheric warming in conjunction with the increase in tropospheric water vapor enhance incoming longwave radiation and thus contribute to surface warming. Surface warming is most evident in winter, although its magnitude and spatial extent depend on the NAG phase. In summer, increases in net shortwave radiation are mainly connected to blocking systems (+NAG) and their drivers are regionally different. In the southern part of Greenland, the atmosphere has become optically thinner due to the decrease in water vapor thus allowing more incoming shortwave radiation to reach the surface. However, we find evidence for southern regions where changes in net longwave radiation balance changes in net shortwave radiation, suggesting the turbulent fluxes control the recent SEB changes. In contrast to South Greenland under +NAG, the moistening of North Greenland has contributed to decreases in surface albedo and enhanced solar radiation absorption. Regardless of the NAG phase, increases in multiple atmospheric variables (e.g., integrated water vapor and net longwave radiation) are found across the northern parts, which suggests atmospheric drivers beyond the heat and moisture originated from the North Atlantic. Especially in the northern ablation zone, sensible heat flux has significantly increased in summer due to larger vertical and horizontal temperature gradients combined with stronger near-surface winds. We attribute the near-surface wind intensification to the emerging open-water feedback, as surface pressure gradients between the ice/snow-covered surface and adjacent open seas are intensified.

# 1 Introduction

The general circulation of the atmosphere over the Greenland Ice Sheet (GrIS) plays a major role in surface melt (e.g., Fettweis et al. 2013; Hanna et al. 2013; Hermann et al. 2020; Tedesco et al. 2016; Tedesco and Fettweis 2020). Given the increasingly strong summer blocking conditions, the air above Greenland has warmed and the GrIS, as well as the peripheral glaciers, have been experiencing mass losses at unprecedented rates since the 1990s (e.g., van den Broeke et al. 2016; Shepherd et al. 2020). Recent studies explain that part of the decreased surface mass balance (SMB) occurs due to snowpack pore saturation in the high-elevation interior (MacFerrin et al., 2019) and in peripheral glaciers (Noël et al., 2017), which has led to less refreezing and thus to enhanced surface meltwater runoff. Moreover, the GrIS albedo feedback (Box et al., 2012) is related to regional increases in rainfall (Bintanja 2018, Niwano et al. 2021), which along prolonged periods with decreased snowfall promotes snow grain aggregation and consequently more solar radiation absorption (Lewis et al. 2021, Noël et al. 2015), thus leading to the migration of the snowline to higher elevations and exposing bare ice (e.g., Noël et al. 2019; Ryan et al. 2019). Recent studies that have investigated the role of temperature inversions over the GrIS report that inversions can effectively trap the near-surface moisture (e.g., Niwano et al. 2019; Shahi et al. 2020) and limit accumulation due to reduced tropospheric mixing (Berkelhammer et al., 2016). Both factors can lead to enhanced surface meltwater runoff, particularly at elevated regions of the GrIS. The surface energy budget (SEB) may also be impacted by changes in the atmospheric lapse-rate given the decreasing surface elevation and the observed high air temperatures (e.g., Gregory et al. 2020; Wang et al. 2021).

According to Ruprich-Robert et al. (2017), the North Atlantic sea surface temperature (SST) experienced a cold period from the mid-1960s until the early 1990s, and since then has warmed at a relatively high rate, mainly due to external climate forcing (e.g., solar, volcanic and anthropogenic). In the last two decades, we have experienced the most intense positive phase of the Atlantic Multidecadal Variability (+AMV, Cassou et al. 2018), with the highest SST anomalies over the North Atlantic since the late 1930s. Particularly during the cold seasons, the literature implies that ocean-atmosphere interaction impacts the jet stream strength, where +AMV leads to a higher frequency of blocking episodes in the North Atlantic (e.g., Athanasiadis et al. 2020; Davini et al. 2015). In recent decades, Greenland blocking has been more persistent and extreme, and has increased notably in winter and summer (Barrett et al., 2020). Extreme Greenland blocking not only leads to relatively warm air advection towards Greenland, it also drags warm and saline Atlantic waters poleward, which then reduces new sea ice-formation across the Greenland Sea (Chatterjee et al., 2021) and in the Baffin Bay (Myers et al., 2021). Some studies report that the direct impact of decreased sea ice concentration may remain confined to the coastal parts of Greenland (e.g., Pedersen and Christensen 2019; Ballinger et al. 2021). However, due to declining sea ice, there has also been an increase in the frequency and intensity of cyclones moving poleward (Valkonen et al., 2021), allowing moisture intrusions to enhance rain/surface melt throughout the year at elevated regions (Oltmanns et al., 2019).

The GrIS is commonly found north of the jet stream, with the North Atlantic storm track to the south. The North Atlantic storm track is more active during cold seasons when baroclinicity is strongest. The resulting cyclonic behavior then favor surface mass gains in South and East Greenland, whereas planetary wave breaks in the North Atlantic generally contribute to surface mass gains through the western part of the GrIS by advecting anomalously warm and moist air polewards (e.g., Liu

and Barnes 2015; Woollings et al. 2008). Such mechanisms in the North Atlantic often form a high-pressure system in the middle troposphere in the vicinity of Greenland depicting the Greenland blocking (Hanna et al. 2015; Woollings et al. 2008). Both cyclones and blocks are essential for the year-round poleward transport of heat and moisture, although the associated thermodynamic and regional impacts vary seasonally (Papritz et al., 2022). Major climate oscillations, such as the North Atlantic Oscillation (NAO) index and the Greenland Blocking Index (GBI), are commonly used to describe the jet stream variability and the predominant atmospheric circulation pattern, a pattern which also influences the variability of the ice sheet mass change. NAO is based on the surface pressure difference between the semi-permanent Subtropical (Azores) High and the semi-permanent Subpolar (Icelandic) Low (Hurrell et al., 2003), and its sign and magnitude provide insight into the North Atlantic jet stream intensity. The NAO phase affects the location and strength of the poleward heat and moisture transport by shaping temperature and precipitation anomalies around the GrIS (e.g., Bjørk et al. 2018; Liu and Barnes 2015; Papritz et al. 2022). GBI describes the mean geopotential height at 500 hPa over Greenland (Hanna et al., 2016). Its index denotes the predominant atmospheric circulation pattern in the vicinity of Greenland, and it regionally governs the heat and moisture transport towards the GrIS interior. The resulting reduction in the equator-to-pole temperature gradient in summer leads to a weakening of the jet stream which then migrates poleward and the atmosphere becomes rather barotropic. Hence, GBI correlates particularly well in summer with near-surface variables and with SMB over the GrIS (Hanna et al., 2013), and since GBI is partially composed of the pressure anomalies over the Northeast Atlantic it is highly correlated with the NAO (Hanna et al., 2015).

The vertical tilt of temperature and pressure within large-scale systems exists due to baroclinicity and recently has been pointed out by Martineau et al. (2020) as an essential mechanism in the North Atlantic for large-scale system development. Therefore, we hypothesize that the tilt within large-scale structures plays a role when calculating climate oscillations, which rely on one parameter at one specific atmospheric level (typically either at 500 hPa or at the surface). We thus suppose that composites of atmospheric and glaciological variables are intrinsically dependent on phase of the concurrent climate oscillation. Particularly in the cold season and under strong cyclonic influence (e.g., atmospheric rivers), the usage of a classification that combines NAO and GBI rather than an isolated one may help to account for specific air mass properties at different atmospheric levels. To overcome this dependency on one atmospheric index, we apply k-means clustering to derive the NAG by using NAO, GBI and the atmospheric water vapor (IWV) over the GrIS. Therefore, NAG links the role of the NAO with the prevailing mid-tropospheric circulation pattern over Greenland (GBI), along with the IWV over the GrIS. Since the NAG estimates the influence of large-scale systems from the North Atlantic on GrIS SEB components, we regionally investigate the climatology of atmospheric variables contributing to SEB for contrasting NAG phases. Finally, we examine changes of SEB components within NAG by comparing recent decades (1991–2020) to a historical period (1959-1990), with a special focus on the summer ablation zone. Section 2 describes the data analyzed, explains the clustering method, and justifies the breakpoint used. Section 3 is broken into three subsections. In Section 3.1 we present the inter-annual variability of the NAG and compare it with NAO and GBI alone; in Section 3.2 we describe the inter-seasonal and regional variability of the NAG; in Section 3.3 we study spatio-temporal anomalies within the same NAG phase, and finally, we concentrate our discussion on regional changes in the summer ablation zone.

## 2 Data and Methods

### 2.1 RACMO2.3p2

The Regional Atmospheric Climate Model (RACMO) was developed and is maintained by the Royal Netherlands Meteorological Institute (KNMI, van Meijgaard et al. 2008). The polar version RACMO2.3p2 is based on KNMI RACMO2.3 but was developed at the Institute for Marine and Atmospheric research Utrecht (IMAU) with dedicated snow physics (Noël et al. 2018, 2019), and was specifically adapted to model the SMB of glacier-covered areas. RACMO2.3p2 is a coupled model (atmospheric and multilayer snow model) that represents meltwater percolation, retention, refreezing and runoff (Ettema et al., 2010). The model combines the dynamical core of the High-Resolution Limited Area Model (HIRLAM) numerical weather prediction model with the ECMWF IFS cycle CY33r1 (Noël et al., 2019). The ECMWF reanalyses products - ERA40 (Uppala et al., 2005) (1959–1978); ERA-I (1979–1989); and ERA5 (1990–2020) - are used to laterally force the atmospheric model (temperature, specific humidity, pressure, wind speed and direction) with additional input of sea surface temperature and sea ice cover within the model domain.

The broadband albedo is calculated as dependent on snow grain radius, solar zenith angle, cloud cover and impurities (soot) concentration in the snowpack (Van Angelen et al., 2012). The bare ice albedo is estimated as the $5^{th}$ percentile of the recorded albedo in each year by the 16-day MODIS product (MCD43A3) over the period 2000–2015. The resulting annual maps of MODIS-derived bare ice albedo are then averaged over the period 2000–2015. In order to better estimate surface mass changes in rugged ablation zones, and in disconnected peripheral glaciers, the original RACMO2.3p2 SMB components at 5.5 km spatial resolution were statistically downscaled to 1 km grid by correcting surface elevation and bare ice albedo biases (Noël et al. 2018, 2019).

RACMO2.3p2, hereafter RACMO2, has been used for many applications over the GrIS with a special focus on surface-atmosphere interaction (e.g., Huai et al. 2020; Lenaerts et al. 2020; Mankoff et al. 2020; Ryan et al. 2019). Recently, the same model version, as part of the GrIS SMB model intercomparison project (GrSMBMIP, Fettweis et al. 2020) was found to provide a realistic representation of the contemporary SMB in the accumulation and ablation zones of the GrIS. Furthermore, Shepherd et al. (2020) and Zou et al. (2020) acknowledged the use of RACMO2 as a complementary tool in estimating GrIS mass changes using satellite data such as GRACE.

### 2.2 Quantifying surface ablation

We have used seasonal (DJF: winter; MAM: spring; JJA: summer; SON: autumn) statistics, in order to characterize the prevailing state of the atmosphere and to avoid potential time lags on the near-surface response due to the heat and moisture transport associated with extreme atmospheric circulation patterns (Barrett et al., 2020) and due to the impact of open water thermal inertia (Hahn et al., 2021; Reusen et al., 2019). Atmospheric variables and SEB fluxes were seasonally averaged, while SMB fluxes were seasonally summed. The area averaged for atmospheric variables varies inter-annually and inter-seasonally depending on the extent of mass gain (accumulation zone) and mass loss (ablation zone) over the GrIS and on the peripheral glaciers (Fig. S1).

The energy available for melt (M) was calculated as:

$$M = SW \downarrow + SW \uparrow + LW \downarrow + LW \uparrow + SHF + LHF + GHF$$

$$= SW_{net} + LW_{net} + SHF + LHF + GHF \tag{1}$$

Downward and upward short/longwave (SW/LW) fluxes are represented by arrows. SHF, LHF and GHF are the sensible, latent and ground heat fluxes, respectively. All terms are in W m$^{-2}$, and represent the snow/ice surface and the energy fluxes received (emitted) by the snowpack are defined as positive (negative). The seasonal surface broadband albedo is the absolute ratio of average SW$\uparrow$ to SW$\downarrow$.

In order to assess if changes in atmospheric variables over land are similar to changes over the adjacent seas (light blue shading in Fig. S1), we divided the adjacent seas into four sectors (delimited by gray lines in Fig. S1): Greenland Sea (Northeast); Iceland/Irminger Sea (Southeast); Labrador Sea (Southwest); and Baffin Bay (Northwest).

## 2.3 Surface ablation trends and break point detection

The literature agrees that the pronounced Greenland summer mass loss started in the 1990s (e.g., Mouginot et al. 2019; Hanna et al. 2021; Shepherd et al. 2020). However, the onset of a clear negative trend varies depending on the time period of each study and on the dataset used. In order to determine the breakpoint of the marked summer surface mass loss in RACMO2, we divided the GrIS into its main seven drainage basins (see Fig. S1) and regionally calculate 612 trends of the summer surface integrated ablation rate for periods with different lengths. For the 62 years of data, the length of sub-periods ranges from 15 (30-year period) to 31 years (62-year period). This will allow the investigation of atmospheric and glaciological conditions prior and post a potential breakpoint. The breakpoint was determined by assessing the most regionally frequent and the largest absolute trend ratios. One trend ratio (RT) is based upon two slopes from equally-sized sub-periods that are split in a common central year. RT is defined as the absolute value of the division between the slope after the central year ($s_2$) and the slope before the central year ($s_1$). For instance (central panel in Fig. 1), $s_1$ between 1977 and 1995 and $s_2$ between 1995 and 2013, whose central year is 1995 gives RT > 1. This means that $s_2$ is more pronounced than $s_1$.

The non-parametric Mann-Kendall (M-K) trend test is used to assess trend monotonicity and significance on summer surface ablation rates (c.f. Section 2.2). The slope corresponds to the Theil-Sen (T-S) estimator. The T-S estimator is a robust regression method that does not require the data to be normally distributed and is hence less vulnerable to outliers than conventional methods. One specific period is considered significant only when the confidence level from the M-K test is higher than (or equal to) 90 % in both sub-periods. Trends in periods exhibiting confidence levels lower than 90 % may still be identical to those exhibiting great significance levels but given their high variability they were not considered. The resulting combination of increasing (i) or decreasing (d) sub-period slopes is shown by color-coded cells (Fig. S2), whereas RTs are displayed in Fig. 1.

The central splitting year and the length of periods with significant sub-period trends for surface mass loss in summer over the GrIS vary regionally. Significant trends are generally detected for sub-periods with lengths between 15 and 25 years. Sub-periods of close to 30 years only occur for regions in the south and west Greenland. The southeast is the only region where

significant trends can be found centered in the mid-1980s. In contrast, the most significant trends in the northern regions are found centered in the mid-1990s. Interestingly, there is only one period in the central-west GrIS with a change in the trend signal (Fig. S2), whereas all others show decreasing trends in both sub-periods.

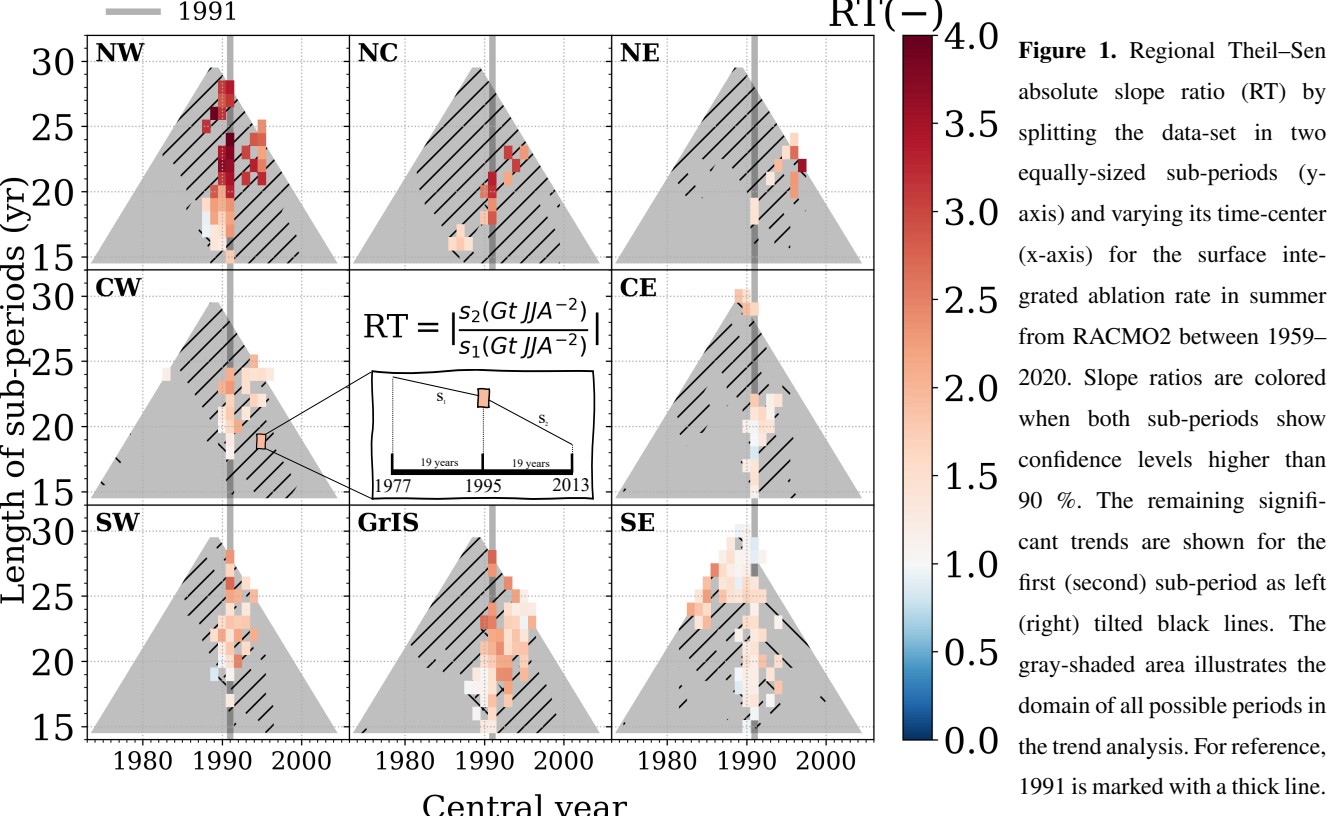

**Figure 1.** Regional Theil–Sen absolute slope ratio (RT) by splitting the data-set in two equally-sized sub-periods (y-axis) and varying its time-center (x-axis) for the surface integrated ablation rate in summer from RACMO2 between 1959–2020. Slope ratios are colored when both sub-periods show confidence levels higher than 90 %. The remaining significant trends are shown for the first (second) sub-period as left (right) tilted black lines. The gray-shaded area illustrates the domain of all possible periods in the trend analysis. For reference, 1991 is marked with a thick line.

The RTs vary depending on the central splitting year and the period length. Major trend shifts are close to 1991 along several period lengths, and 1991 corresponds to the central splitting year with most significant trends amongst regions. Especially for periods ranging between 40 and 50 years, RTs generally increased in magnitude with latitude. In the northern regions, slopes of summer ablation rates during the second sub-period are four times larger than during the first sub-period, while in the southern regions, more specifically in the southeast, the first sub-period slopes are only occasionally larger (RT < 1).

Based on this assessment, we use 1991 as the year to split the period 1959–2020 into two sub-periods and to explore inter-seasonality of atmospheric and glaciological variables as a function of the prevailing atmospheric circulation. The same year was also used in recent literature (e.g., van den Broeke et al. 2016; Noël et al. 2019; and Hanna et al. 2021) for trend analysis and will hence allow for direct comparison of results.

## 2.4 Combination of climate oscillations

There are several methods that can be used to define the NAO (e.g., principal component analysis or k-means clustering) as well as data (reanalysis or station-based). Also, sea-level pressure (e.g., NCAR/UCAR, Hurrell et al. 2003) or 500 hPa geopotential

height (NCEP/CPC, van den Dool et al. 2000) are typically used within one specific method (e.g., principal component analysis) to calculate NAO. Here, NAO derived from the leading principal component based on sea-level pressure anomalies over the Atlantic sector (20° N–80° N, 90° W–40° E) from NCAR/UCAR (Hurrell et al., 2003) is used. This product is supposed to better represent the full spatial patterns of the NAO than the product based on a specific surface station. It is nevertheless important to highlight that NAO derived from principal component analysis is in constant adjustment with the inclusion of

new data. GBI is derived from 500 hPa geopotential height over the region (60° N–80° N, 80° W–20° W) and is obtained from PSL/ESRL (Hanna et al., 2016). Both climate oscillations originate from NCEP/NCAR reanalysis (Kalnay et al., 1996). Ultimately, both products were seasonally standardized relative to the period 1950–2000.

In order to understand the extent to which NAO and GBI are related to or influenced by other climatic indices, data on the Arctic Oscillation (AO) and on the Atlantic Multidecadal Variability (AMV) were obtained from CPC/NCEP and PSL/ESRL,

respectively, and analyzed for the period 1959–2020. The non-parametric Spearman correlation coefficient ($r_S$) was calculated in order to quantify the relationships (strength and direction) between the variables. Seasonal GBI and NAO are highly and negatively correlated. However, in summer GBI correlates better (-0.8) with Greenland SMB rates than NAO (0.5) with a 99.9 % confidence level. The GBI is also influenced by other atmospheric/oceanic patterns and correlates best with AO in winter (-0.9) while NAO correlates with AO better during the remainder of the year (-0.9), and AMV shows the greatest correlation

with GBI during the summer (0.5). Cross-correlation was applied to the climate indices and the entire GrIS surface mass fluxes in order to assess potential links associated with the near-surface climate triggered by the atmospheric circulation in preceding seasons. However, no substantial improvements in correlation were found. This suggests that there is no relevant time-lag response between seasonal GrIS SMB from RACMO2 and the prevailing atmospheric circulation pattern in preceding seasons.

The composite analysis of one climate oscillation alone may not be sufficient to understand the atmospheric circulation

influence on surface processes caused by the other. In addition, the inclusion of GrIS integrated water vapor (IWV) in one classification can also reinforce the role of the two climate oscillations with respect to heat and moisture advection towards Greenland. To take these requirements into account, we apply k-means clustering to NAO, GBI and GrIS IWV to estimate the "influence" of large-scale systems over the North Atlantic in Greenland and name the derived classification as NAG. According to within-cluster sum of squares, a measure of variability within each cluster, the optimal number of clusters for our

data is not larger than 3. Also, as climate oscillations are commonly identified as positive, neutral and negative phases, three clusters (+NAG, 0NAG, –NAG) were defined in advance. The three seasonal variables considered (NAO, GBI, GrIS IWV) are represented by 62 points/years in a 3-dimensional space. As an initial condition, three random points are selected in space to serve as the center of each cluster. The 3-dimensional Euclidian distances between the 62 points and the center of the three random clusters are calculated. Points are classified individually based on their distance to the center of the closest cluster.

The center of the three clusters shifts iteratively by the mean distances of all points within its own cluster. The best possible

grouping is achieved by selecting the minimum calculated sum of squares of the distances between grouped points and the mean center of each group. The k-means clustering method is then repeated seasonally. The resulting clustering (Fig. S3) is sensitive to the choice of the time period, number of clusters defined and variables. A sensitivity analysis of the clustering and percentile classification using NAO (van den Dool et al., 2000) derived from 500 hPa geopotential height or NAO (Hurrell et al., 2003) derived from surface pressure and GBI is addressed in the Supplementary Material.

# 3 Results and Discussion

## 3.1 The influence of the North Atlantic over Greenland

The large-scale circulation involved in individual NAG phases is shown inter-seasonally in Fig. S4, where the positive phase of NAG is connected to an anomalously high geopotential height at 500 hPa level (GBI > 0) as well as high IWV, and to the anomalously negative pressure difference between the semi-persistent Azores high and the semi-persistent Icelandic low (NAO < 0).

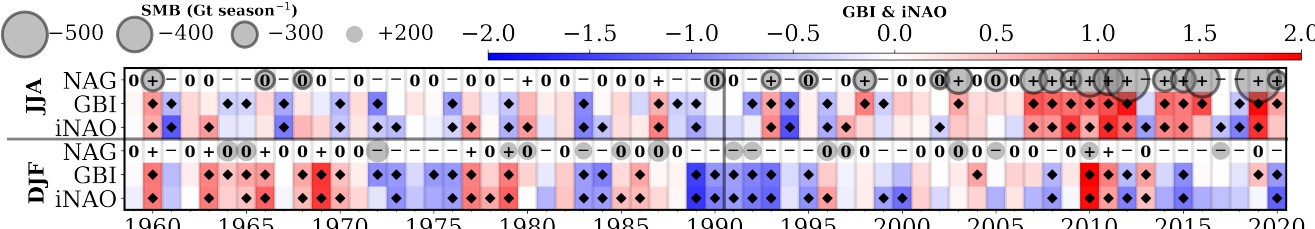

**Figure 2.** Time-series of seasonal GBI, signal inverted NAO (iNAO) and NAG. GBI and iNAO phases are color-coded and NAG is coded by symbols shown positive (+); neutral (0); and negative (–) phase. The negative (positive) GBI and iNAO phase based on the 25[th] (75[th]) percentile are illustrated as diamonds. Seasonally accumulated surface mass balance (SMB) for absolute quantities larger than 200 Gt season$^{-1}$ (winter DJF and summer JJA) is sized accordingly. A negative SMB is marked by a dark circle around the bubble. For reference, 1991 is highlighted as a gray vertical line to illustrate GBI, NAO and NAG phases.

In winter, the center of the high-pressure system (+NAG) is situated over the Baffin Bay as also described by Woollings et al. (2008). In summer +NAG leads to a ridge, stretching from Baffin Bay to North Greenland. Despite the typical life cycle of the NAO phase lasting about two weeks (Feldstein, 2003), the geopotential height vertical tilting described by Martineau et al. (2020) remains within seasonal composites due to strong baroclinicity (Fig. S4). Particularly in winter under –NAG, a well-marked jet stream is present over the North Atlantic that bends and stretches northeast along the Greenland Sea. The inter-annual NAG is shown in Fig. 2, alongside seasonal GBI and signal inverted NAO (iNAO, only here used for qualitative purposes). Absolute quantities of seasonally accumulated SMB larger than 200 Gt season$^{-1}$ are also depicted in Fig. 2. The seasonal SMB indicates surface mass changes over both the GrIS and the peripheral glaciers. Strongest mass losses coincide with summer, and are mostly connected to +NAG after 1991. Whereas in summer and autumn +NAG contributes the least to surface accumulation, in winter +NAG can contribute to high surface accumulation. In spring, opposite NAG phases can contribute equally to seasonal accumulation in the SMB. A more comprehensive view of seasonal and spatial integrated SMB is seen in Fig. S5.

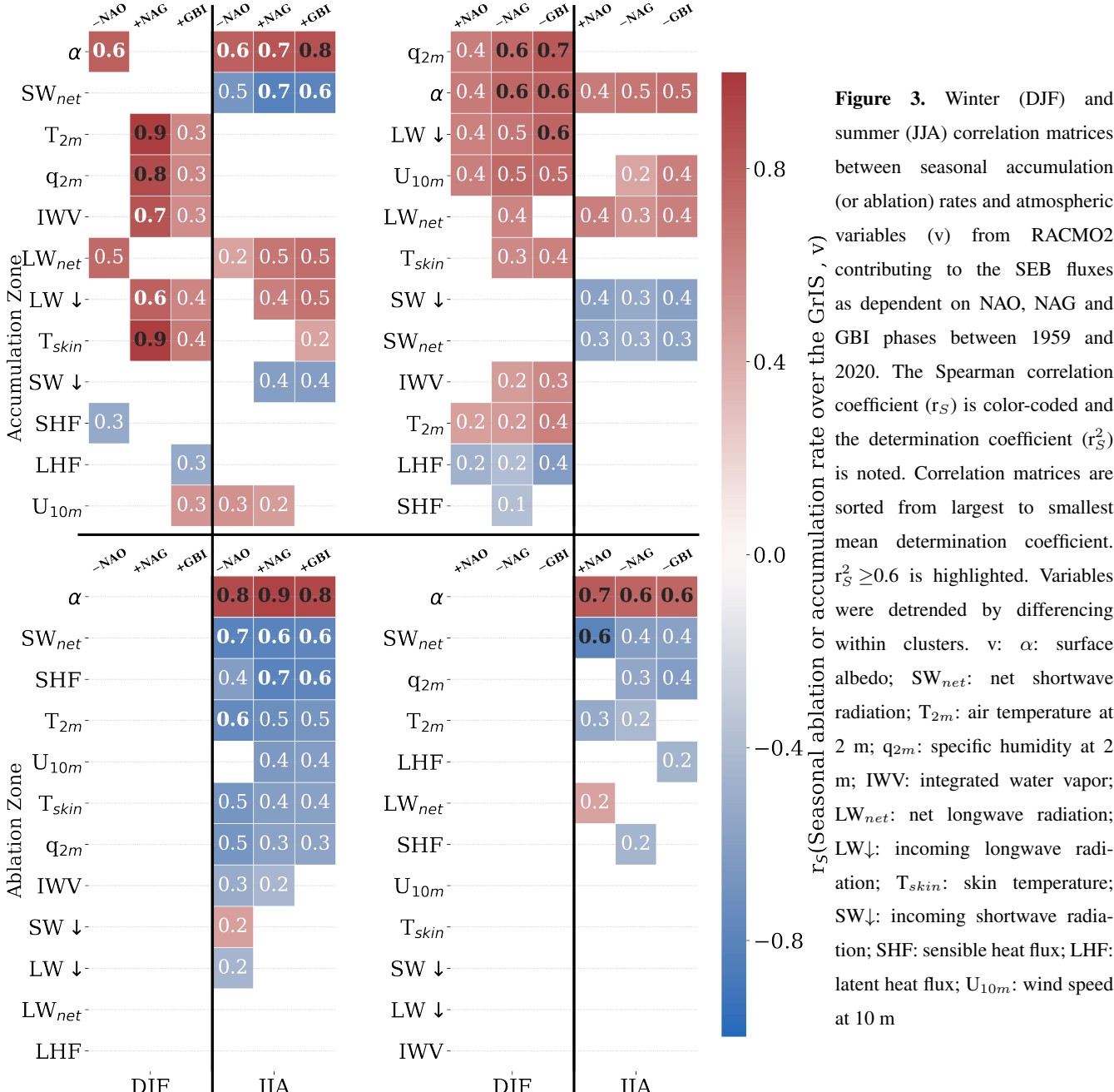

**Figure 3.** Winter (DJF) and summer (JJA) correlation matrices between seasonal accumulation (or ablation) rates and atmospheric variables (v) from RACMO2 contributing to the SEB fluxes as dependent on NAO, NAG and GBI phases between 1959 and 2020. The Spearman correlation coefficient ($r_S$) is color-coded and the determination coefficient ($r_S^2$) is noted. Correlation matrices are sorted from largest to smallest mean determination coefficient. $r_S^2 \geq 0.6$ is highlighted. Variables were detrended by differencing within clusters. v: $\alpha$: surface albedo; $SW_{net}$: net shortwave radiation; $T_{2m}$: air temperature at 2 m; $q_{2m}$: specific humidity at 2 m; IWV: integrated water vapor; $LW_{net}$: net longwave radiation; LW↓: incoming longwave radiation; $T_{skin}$: skin temperature; SW↓: incoming shortwave radiation; SHF: sensible heat flux; LHF: latent heat flux; $U_{10m}$: wind speed at 10 m

Correlations between seasonal accumulation (or ablation) rates and atmospheric variables as dependent on the NAO, NAG and GBI phase are shown in Fig. 3. Under anticyclonic conditions, +NAG shows in winter and summer higher correlations between seasonal SMB rates and atmospheric variables than –NAO and +GBI over the GrIS accumulation zone. In summer, such correlations for the ablation zone are similar for NAO, NAG and GBI. In spring and autumn, +GBI and –NAO generally

show higher correlations between seasonal SMB rates and atmospheric variables than NAG in the ablation zone, except in the accumulation zone where the opposite is found (Fig. S6). Under neutral phases and under strong jet stream conditions, there are relatively small differences among NAO, NAG and GBI concerning the correlations between SMB rates and atmospheric variables.

## 3.2 Inter-seasonal NAG climatology

Spatial and inter-seasonal anomalies under contrasting NAG (+/–) phases with respect to the neutral phase (0NAG) are illustrated in Fig. 4 (and Fig. S7) for IWV, incoming longwave radiation (LW↓), specific humidity at 2 m ($q_{2m}$) and skin temperature ($T_{skin}$). Seasonal $T_{skin}$ and the air temperature at 2 m($T_{2m}$) are highly and positively correlated ($r_S > 0.9$) in the ablation and accumulation zones for contrasting NAG phases. Differences in their correlation are small and only found close to the ice-sheet margins in summer where the snow/ice surface is physically constrained to 273.15 K (not shown). Moreover, increases in seasonal $T_{2m}$ are accompanied by exponential increases in $q_{2m}$. Spatial and inter-seasonal anomalies for wind speed at 10 m ($U_{10m}$), SHF, LHF, and (ice+liquid water) cloud content are shown in Fig. S8.

High IWV occurs mainly along coastal areas and rapidly decreases towards the elevated interior regardless of the NAG phase (Fig. 4a). Major IWV differences are found in all seasons in West Greenland where meridional heat and moisture advection is promoted by +NAG. In winter and summer, the LW↓ signal (Fig. 4b) agrees with IWV anomalies for both phases. However, negative anomalies in LW↓ are not related to negative anomalies in IWV. This occurs because IWV combines the $q_{2m}$ and the remaining water vapor in the lower-troposphere which is typically associated with the cloud content. IWV differs the most from $q_{2m}$ in summer close to the transient equilibrium zone (SMB = 0) where the largest amounts of $q_{2m}$ are found due to the expansion of the melting area under +NAG. This is particularly visible in West Greenland. Also, the flat northeast interior experiences rather high levels of $q_{2m}$ and LW↓ which are as high as those in the ablation area, a consequence of high cloud content that promotes low cloud/fog conditions.

While the liquid water within clouds (LWP) lies mainly along the coastline, the ice content within the clouds (IWP) spreads from the coast further inland, exhibiting opposite patterns in winter and summer: +NAG(–NAG) in winter promotes more IWP at the Northwest (Northeast), while in summer increases (decreases) in IWP are favored over the whole of Greenland under –NAG(+NAG). In spite of the relatively small but highly radiative cloud content in the North (Fig. 4b), the SW↓ is only partly attenuated. Moreover, a small increase in LWP compensates a small decrease in IWP, and hence the cloud content varies little under +NAG relative to 0NAG over the same region (Fig. S8d).

**Figure 4.** Seasonal and spatial anomalies for (a) integrated water vapor (IWV), (b) incoming longwave radiation at the surface (LW↓), (c) specific humidity at 2m ($q_{2m}$) and (d) skin temperature ($T_{skin}$) for opposite NAG phases with respect to the neutral phase ($\overline{+(-)NAG}$ - $\overline{0NAG}$) between 1959 and 2020 from RACMO2. The relative seasonal frequency (f in %) of each NAG phase (+NAG, 0NAG and −NAG) used to produce composites is indicated as subtitle in a). For reference, Summit and South Dome are marked with big and small black triangles, respectively. Stippled regions indicate areas with a confidence level greater than 90 % (based on the Wilcoxon rank-sum statistic test for unpaired sets). See Fig. S7 to examine seasonal and spatial anomalies in spring (MAM) and autumn (SON).

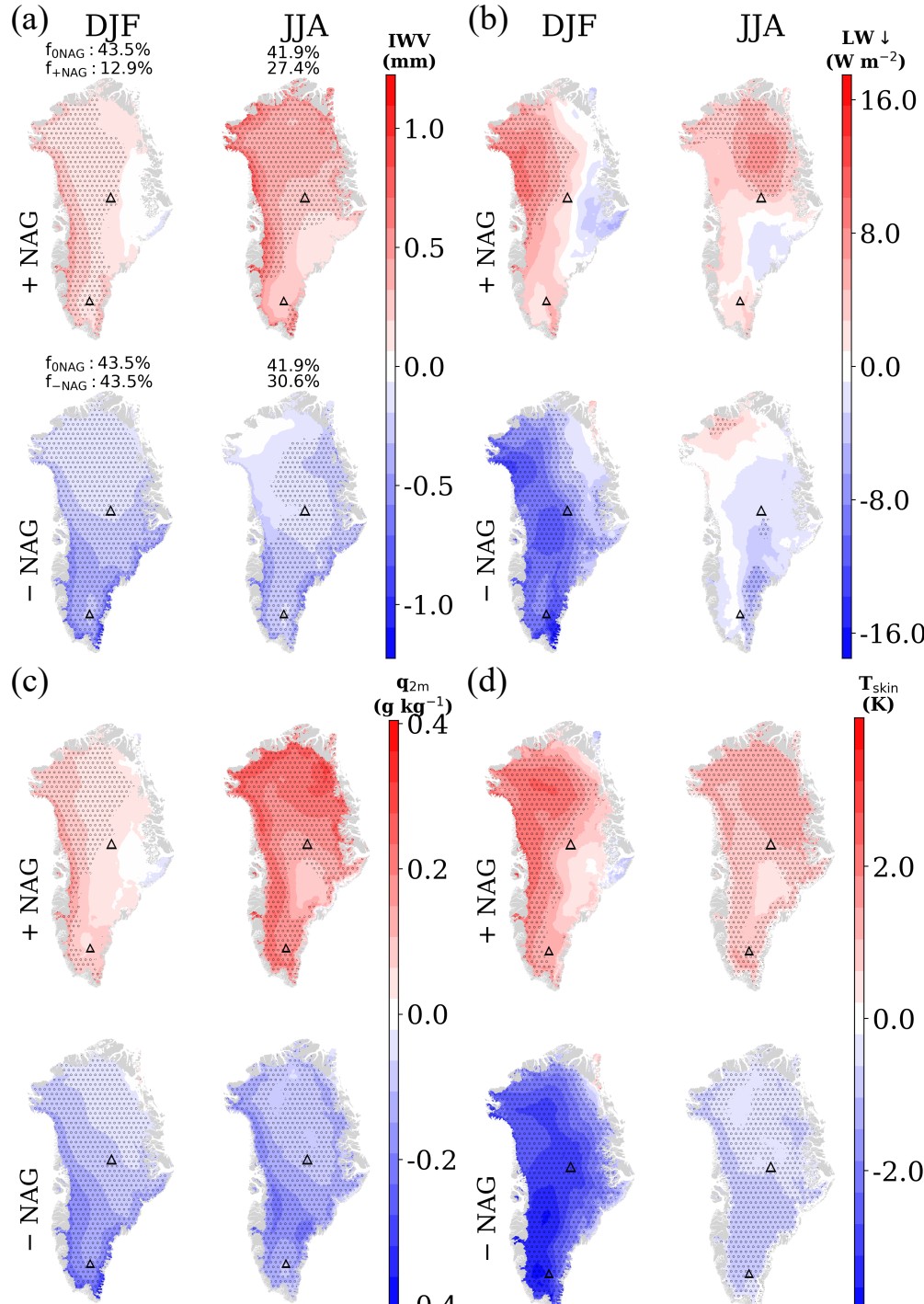

The largest $T_{skin}$ anomalies (Fig. 4d) are found in winter, more specifically in North Greenland, where the +NAG is up to 4 K warmer than 0NAG. Positive anomalies in summer are of lower amplitude in comparison with 0NAG, but significant in the North.

      Particularly across the northern regions, the near-surface winds are stronger in winter during +NAG, whereas under –NAG phase they are weaker (Fig. S8a). These intense near-surface winds are explained by the coupling of the katabatic winds with
265 upper winds and are referred to "Greenland plateau jets" (Moore et al., 2013). Such Greenland plateau jets are known to enhance radiative effects on the surface during melt events (e.g., Mattingly et al. 2020). Under the –NAG, the barrier winds in the southeast become stronger as a result of orography-cyclone interaction, from the Denmark Strait to the southernmost tip of Greenland. Also, due to the steep surface the near-surface winds at this location are strong over the entire year, whereas most of the near-surface flow around the GrIS is weaker with respect to the 0NAG. During the cold seasons, when the near-surface
radiation deficit is greatest, and under +NAG conditions, enhanced near-surface winds also contribute to the surface warming. Particularly in the North GrIS, the strong plateau jets are associated with the high SHF that prevents surface cooling. In general, SHF is mostly driven by the high near-surface wind speeds, with the exception of the southern ablation zone during summer under +NAG, where winds decrease but the near-surface temperature and specific humidity gradient increase (Fig. S8b, c), resulting in positive SHF and LHF anomalies with respect to 0NAG. In contrast, the near-surface wind under the +NAG phase
redistributes the near-surface water vapor and reduces the LHF for most of the year. Except in the southeast, the LHF becomes less negative under –NAG due to a weakening of the katabatic flow all over the GrIS.

## 3.3    Inter-seasonal NAG variability

### 3.3.1    Seasonal and spatial anomalies

Figure 5 shows the seasonal and spatial anomalies depending on the NAG phase (see 0NAG in Fig. S9) for the period 1991–
280 2020 based on the reference period (1959–1990). The respective composite for separated GBI and NAO are shown in Fig. S10 and S11 according to a seasonal percentile classification (see Fig. S3 to compare seasonal classifications). The fraction of years (f) before and after 1991 within each composite is stated at the top of the Fig. 5a. An unchanged surface-atmosphere interaction under a specific atmospheric circulation pattern would lead to anomalies close to zero. However, we find significantly positive deviations over the GrIS and adjacent seas in most seasons for all NAG phases.

The amount of water vapor in the atmosphere (Fig. 5a) has notably increased, especially in North Greenland (Fig. S7a). Significant increases of IWV are also found in winter along West Greenland due to increases in cloud content, partly related to increases of IWP. Especially during cold seasons, –NAG phases exhibit general atmospheric warming along the coast. Moreover, given the increase in cyclonic activity in recent decades along the Fram Strait, more heat and moisture is potentially advected toward Northeast Greenland under –NAG. The combined warm and moist air has also enhanced LW↓ in the region.
In the interior, the relative warm but dry atmosphere explains the increase in LW↓.

      In summer, moisture increased poleward in all NAG phases. The major significant increases in IWV during summer occur in the north under +NAG and along the west coast under 0NAG (Fig. S7a). Over the adjacent seas, the largest anomaly in IWV

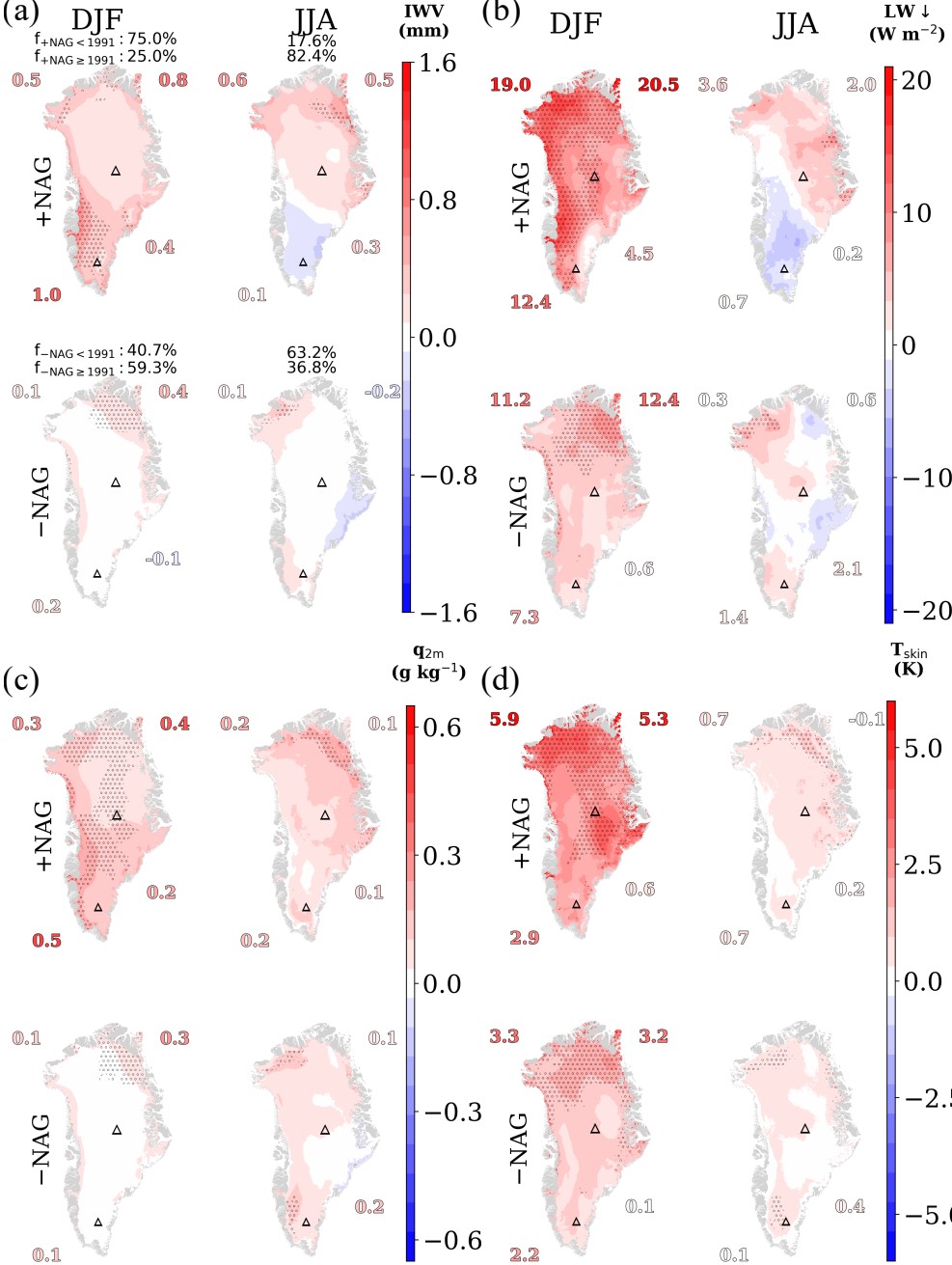

**Figure 5.** Seasonal and spatial anomalies for (a) integrated water vapor, (b) incoming longwave radiation reaching the surface, (c) near-surface specific humidity, and (d) skin temperature from RACMO2 between 1991–2020 and 1959–1990 as dependent on the NAG phase. The percentage (f) of the NAG phase in each period is indicated for each season. For reference, Summit and South Dome are marked as big and small triangles, respectively. Stippled regions indicate areas with a confidence level greater than 90 % (based on the Wilcoxon rank-sum statistic test for unpaired sets). Temporal anomalies between composites over the adjacent seas are also shown as colored numbers (Baffin Bay: upper left; Greenland Sea: upper right; Irminger Sea (lower right) and Labrador Sea (lower left). Temporal anomalies equal to null are omitted. See Fig. S1 to discern the extension overseas and Fig. S9 to examine seasonal and spatial anomalies under 0NAG.

(nearly 1 mm) is found in autumn under –NAG in the northeast (Fig. S9). Also, similar IWV increases in winter during +NAG are found in the northeast and southwest. Increases in IWV over the adjacent seas are essentially related to surface heating due to sea ice decrease in recent decades.

The decreasing summer cloud cover trend in the southwest reported by Lim et al. (2016) and Hofer et al. (2017) is consistent with the negative anomaly in IWV found under +NAG. Decreases in water vapor in the atmosphere extend until autumn (Fig. S9). However, these are not significantly different from those in the reference period under the same atmospheric configuration. An increase in $SW_{net}$ is found over the same southern regions as a consequence of more incoming SW radiation. Since the adjacent southern seas of Greenland show an opposite change, the inland decreased IWV has to be driven by regional effects. The light winds, the incoming SW radiation and $q_{2m}$ increase at the surface point to subsidence in the region. As a result of surface inversions favored by subsidence in association with surface melt, southern regions comprise more $q_{2m}$ for the period 1991–2020 than during the reference period (Fig. 5c). These findings are in line with Niwano et al. (2019), who also pointed to the importance of the latent heat released by near-surface water vapor for ablation processes in the region.

Despite the SW↓ (Fig. S12b) and the cloud water content little changed over the northern regions under +NAG, $SW_{net}$ (Fig. S14) significantly increased mainly due to lower surface albedo. Noël et al. (2019) attributed the recent decrease in surface albedo over the northern regions to rising atmospheric temperatures and increased cloudiness. The LWP increase, which also contributes to the increase in LW↓ (Fig. 5b), is particularly pronounced in the Northwest regardless of the NAG phase. Other regional factors such as the increase in SHF and wind speed could have also contributed to the surface albedo decrease.

The largest $T_{skin}$ anomaly is found in winter under +NAG in North Greenland, where a similar temperature increase (nearly 5 K) is also found over the adjacent northern seas. For other NAG phases and regions, $T_{skin}$ anomalies over land are larger than over the adjacent seas. Except in the southeast where the SHF explains the near-surface warming, the $T_{skin}$ positive anomaly during winter is mainly a result of the low-tropospheric warming (Fig 5b, d). Contributions to surface warming due to more water vapor in the atmosphere in winter are particularly strong under +NAG and otherwise confined to southern parts and coastal areas. This $T_{skin}$ warming in recent decades has the potential to affect the snow metamorphosis regionally, and hence a thinner snow layer can more efficiently and more quickly expose darker layers in the following summer regardless of the prevailing atmospheric circulation pattern and hence accelerates regional surface mass loss. Nevertheless, the NAG phase in summer governs the overall surface mass loss. For all NAG phases, summer shows the lowest $T_{skin}$ anomalies, although significant differences in $T_{skin}$ are found in the north (Fig. 5d). In other words, the north of Greenland warmed in all NAG phases. $T_{skin}$ anomalies extend to the west of Greenland and to the entire south under 0NAG (Fig. S9d). Despite the fact that the $T_{skin}$ is physically limited to 273.15 K, wide areas show an increase in $T_{skin}$, which is in line with proportional increases in $T_{2m}$. With respect to the reference period, spring under –NAG is found to exhibit surface cooling in recent decades. However, the opposite is found for the northern GrIS and over adjacent seas under other NAG phases.

Figure S10 and S11 show spatial anomalies for GBI and NAO percentile classifications, respectively. In spite of the fact that GBI and NAO are highly negatively correlated in summer, they show distinct results when categorized and analyzed separately. This suggests temporal changes in one climate oscillation index that are not accounted for by the other, and vice versa. The exceptionally high summer GBI values in recent decades (e.g., Barrett et al. 2020; Hanna et al. 2016; Hanna et al. 2018)

have led to an increase in the 1959–2020 percentile threshold, and thus prevents the detection of similar index magnitudes before 1991. Also, Wachowicz et al. (2021) have recently pointed to GBI inconsistencies due to the amplified warming at high latitudes. The same explanation holds for the rest of the year, as the NAO-GBI correlation is relatively weak.

Whereas decreases in IWV and $q_{2m}$ are found in summer across the southern regions under +GBI, IWV and $q_{2m}$ values under –NAO in contrast exhibit increase compared to the reference period (1959–1990). This illustrates the crucial role of NAO advecting heat and moisture through storms migrating poleward toward Greenland, an effect that is not fully captured by GBI at the 75$^{th}$ percentile threshold. In spite of the increasing IWV in summer during –NAO, small increases in SW↓ are also found under +GBI and –NAO over Northeast Greenland. During winter, all variables in our analysis show larger differences for NAO than for GBI. Particularly for IWV and $q_{2m}$, only relatively small differences are found for GBI and significant ones are only found under NAO. However, both climate oscillations register positive anomalies in LW↓ and $T_{skin}$, a consequence of the general atmospheric warming. Major surface warming is found under NAO, reaching anomalies higher than 5 K over the GrIS interior, whereas the highest temperature increase for GBI anomalies is found in autumn under the negative phase. Warming anomalies over the adjacent seas are typically larger under –NAO, but over the northern GrIS the warming is similar for contrasting NAO phases. In fact, positive anomalies in multiple atmospheric variables are found across the northern parts regardless of the climate oscillation or phase, which suggests impacts of atmospheric drivers beyond the prevailing atmospheric circulation over the North Atlantic. Both climate oscillations (–GBI and +NAO) are in agreement with –NAG concerning the anomalously cold and dry spring.

### 3.3.2 SEB changes in the summer ablation zone

Figure 6 shows regional changes for surface energy components in the summer ablation zone as dependent on the NAG phase. Regional changes in $T_{2m}$, $T_{skin}$, near-surface temperature gradient ($\Delta T = T_{2m} - T_{skin}$), IWV, $U_{10m}$ and individual radiation fluxes are shown in Fig. S12. Most regions show an increase in SHF for the period 1991–2020 regardless of the NAG phase. The SHF increase is linked to the intensification of the near-surface winds and the strengthening of $\Delta T$ (Fig. S12a). As the temperature of the melting snow/ice surface is physically limited to 273.15 K, the $\Delta T$ is essentially driven by the air temperature increase. The melting snow/ice surface, in conjunction with steep slopes, promotes downslope winds. In addition, contributions to the marked wind speed strengthening in the ablation zone arises from the migration of the snowline to higher elevations of the ice-sheet (Ryan et al., 2019), which in turn enhances the surface pressure gradient and adds momentum to the flow. Particularly in the northern regions, one factor that contributes to the increase in summer wind speed is the decrease in ice in the neighboring seas. The change in wind speed can be related to emerging open water feedback as it occurs irrespective of the prevailing atmospheric circulation pattern. The increase in wind speed favors polynya formation that generates low surface pressure over the open waters and hence enhances the regional surface pressure gradient. A thermal circulation is identified over the North and Northeast Greenland, where margins are almost permanently ice-covered during the reference period. In the Northwest ablation zone, the wind speed has not increased significantly potentially related to the seasonal Baffin Bay ice-free.

$SW_{net}$ has also increased in both the northern and southern regions, and is commonly associated with +NAG. Whereas in the south $SW_{net}$ increased due to more SW↓ as a result of an optically thin atmosphere (Hofer et al., 2017; Lim et al., 2016), in the

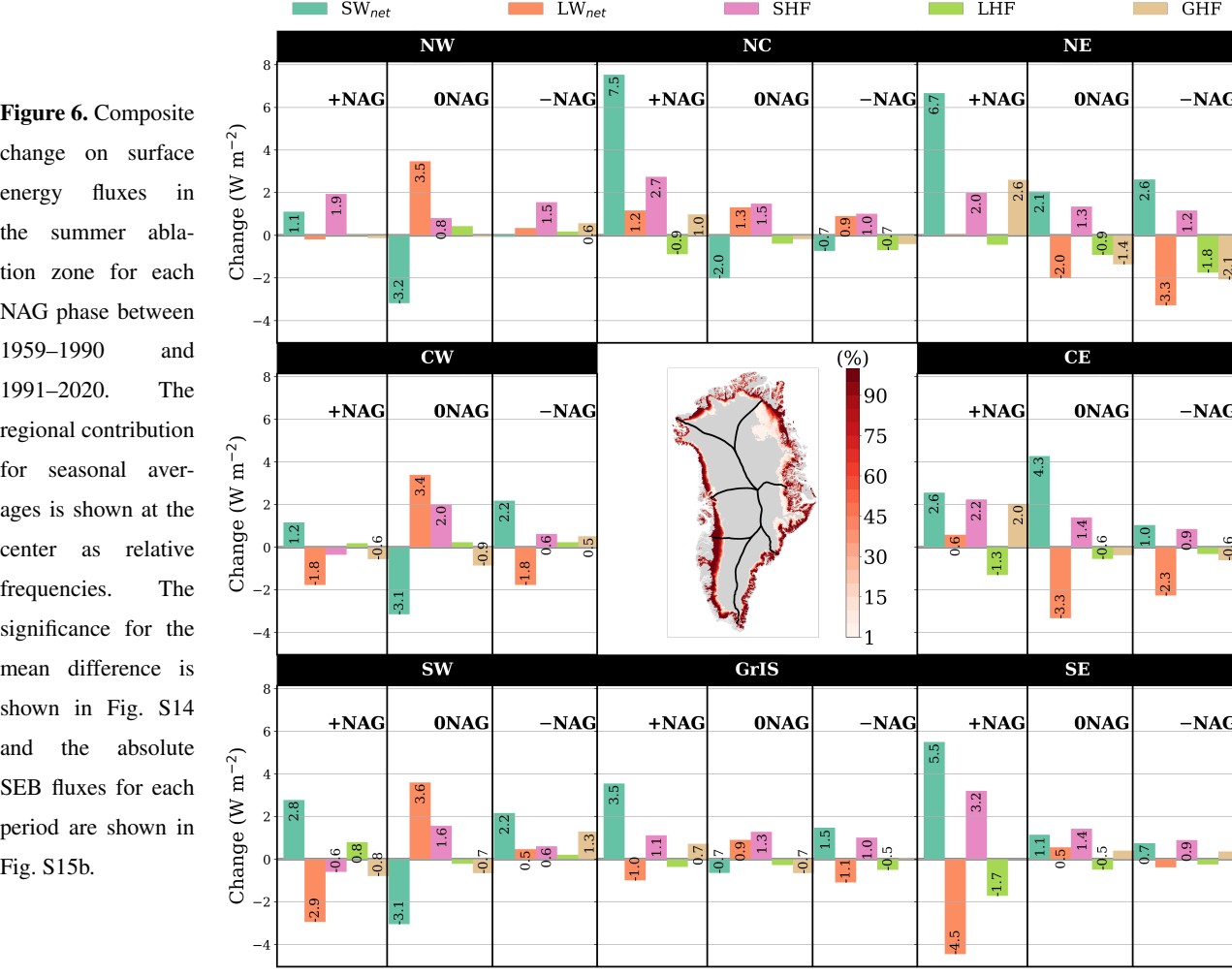

**Figure 6.** Composite change on surface energy fluxes in the summer ablation zone for each NAG phase between 1959–1990 and 1991–2020. The regional contribution for seasonal averages is shown at the center as relative frequencies. The significance for the mean difference is shown in Fig. S14 and the absolute SEB fluxes for each period are shown in Fig. S15b.

north the increase is due to a darker surface as a consequence of the expansion of the bare ice area (Noël et al., 2019). In other words, net solar radiation changes in the south are accompanied by decreases in IWV, whereas in the north, they are associated with decreases in SW↑ (Fig. S12b). SHF and $SW_{net}$ are the largest changes in SEB fluxes in most regions regardless of the

365 NAG phase. These two SEB components have recently been reported as being the main melt drivers in recent decades (Wang et al., 2021). As the summer ablation zone emits in both periods close to melting point (nearly 315 $Wm^{-2}$), the reduced $LW_{net}$ is driven by the reduced LW↓. Interestingly, changes in $LW_{net}$ regionally outweigh changes in $SW_{net}$, making the turbulent fluxes control SEB changes.

Apart from a few isolated areas in the north under +NAG, there is not a strong signal in GHF. Particularly in the southeast,

GHF used to contribute more to SEB than SHF, but in recent decades the role of SHF on SEB has become higher than that of GHF (Fig. S15b). In spite of the wind speed increase over large areas, we find small differences in LHF (< 1 W $m^{-2}$) under +NAG. In contrast to the air temperature and the $q_{2m}$, the summer $RH_{2m}$ (on average higher than 75 % regardless of the NAG

phase) changed little during summer over the GrIS. Particularly in the northern regions under +NAG, as the atmosphere has become warmer and moister, less latent heat is consumed to maintain the levels of moisture above the surface. Thus, the vertical mixing of moisture due to stronger winds has potentially become less efficient. This is true for most of the northern ablation zone, but not for South Greenland. In the lowest part of the GrIS ablation zone and peripheral glaciers, the $RH_{2m}$ decreased and so did the LHF. Cancelling out positive and negative changes within the ablation zone could have also affected the LHF spatial average. The largest changes in temperature, IWV, wind speed and energy available for melt in the summer ablation zone occurred under 0NAG, especially over North and East Greenland (Fig. S12a).

Surface melt occurs sporadically in the summer accumulation zone. The summer accumulation zone has been decreasing in area as a result of the upward migration of the snowline (Noël et al., 2019). The air temperature increase is one of the largest changes occurring in the summer accumulation zone, irrespective of the NAG phase (Fig. S13). This favors a decrease in surface albedo, more surface absorption of solar radiation and possibly an increase in the frequency of melt events. Specifically under +NAG, the optical thinning of the atmosphere allows enhanced SW↓ to warm and to darken the surface. As in the ablation zone, and regardless of the NAG phase, SHF has increased in most regions. However, here it is additionally accompanied by considerable increases in $SW_{net}$ and decreases in $LW_{net}$. $LW_{net}$ decreases in the south are related to similar LW↑ fluxes in both periods, while the $LW_{net}$ increases in the north are associated with more water vapor in the atmosphere, which then enhances LW↓.

In winter, most of the accumulation zone has warmed with respect to 1959–1990 (Fig. S17). The resulting warming bears more water vapor near the surface and at elevated levels of the lower-troposphere over Greenland. Independent of the NAG phase, LW↓ is particularly larger than LW↑ in West Greenland due to a warmer and more humid atmosphere. Consequently, SHF has decreased in identical magnitude over the same regions (Fig. S16a). Under +NAG, the opposite changes occurred in the southeast, especially due to the strengthening of the wind. Except in North Greenland, overall temperature increases are dependent on the NAG phase.

## 4 Conclusions

Using output of a regional climate model over the GrIS and adjacent seas, 62 years (1959–2020) of climate variability were analyzed at inter-seasonal scale. A clustering method enabled the Greenland Blocking Index (GBI) and the North Atlantic Oscillation (NAO) to be combined with the integrated water vapor (IWV) in order to derive the North Atlantic influence on Greenland (NAG). This approach was used for the first time to link pronounced atmospheric blocking conditions over Greenland and the GrIS with pronounced surface pressure gradients in the North Atlantic to describe climate variability. Given the importance of poleward moisture transport on the surface energy fluxes, IWV was also included in the cluster analysis. This helped to better separate neutral climate oscillation phases from the rest (since NAO and GBI are often linearly related, and their classification is ambiguous if close to zero). The resulting clustering allowed for characterization of atmospheric circulation patterns that capture the variable influence of the North Atlantic on Greenland. NAG differs from classifications based on seasonal percentile thresholds. Moreover, typical climate features marked under certain atmospheric circulation patterns in

certain seasons were possible to describe. Also, NAG proves its value for not depending solely on one sensitive climate oscillation index, and therefore agglomerates similar NAO, GBI and IWV conditions relevant for Greenland that are not captured by isolated indices.

Inter-seasonal NAG anomalies show strong effects for West Greenland in comparison to the neutral phase, but there are also marked anomalies over the entire GrIS. Larger inter-seasonal differences were found in winter and summer, particularly affecting northern regions. Regional anomalies are found in recent decades compared to the reference period (1959–1990) for the three NAG phases. The magnitude of these anomalies depend on season and NAG phase. Particularly along the coastline, increased air temperature in winter allows for more water vapor in the air, albeit without necessarily resulting in saturation. The enhanced atmospheric warming is more pronounced for the period 1991–2020 under +NAG. We attribute the increase in near-surface specific humidity and the general tropospheric warming to strongly drive surface heating through enhanced downward longwave radiation (LW↓) in winter. Surface warming is particularly marked over North Greenland and over the adjacent seas among NAG phases. However, the vertical distribution of changes in the lower troposphere, i.e., temperature and water vapor changes associated with temperature and humidity inversions and with cloud phases, require further investigation to assess their contribution to surface warming.

The factors that have contributed to a higher SEB over the GrIS in recent decades vary across accumulation and ablation zones. The increase of SHF occurs in both zones due to stronger winds and higher temperatures near the surface. However, particularly in the ablation zone, increases in SHF are similar for contrasting atmospheric circulation patterns, suggesting the influence of drivers beyond the prevailing atmospheric circulation pattern. For example, the increased surface pressure gradients between the ablation zones and the adjacent seas, suggests that the decline of sea ice in recent decades is one such driver, manifesting as the emerging open-water feedback principally in North Greenland. However, more investigation is needed to ascertain the specific role of sea ice concentration in the neighboring seas. An analysis at a higher temporal resolution and of extreme events, would also shed light on the factors behind atmospheric changes and surface melt drivers.

The optical thinning of the lower-troposphere, a characteristic found mainly under +NAG, results in enhanced incoming shortwave radiation, especially over South and East Greenland. However, changes in net shortwave radiation balance changes in net longwave radiation in South Greenland, highlighting the importance of changes in turbulent fluxes. Prolonged warm periods without fresh snow can nevertheless contribute to the darkening of the surface and consequent upward migration of the snowline. In contrast, summer increases in the atmospheric water vapor over the northern accumulation zone are independent of the NAG phase. Despite the water vapor increase in recent decades, there have been relatively little changes in the incoming shortwave radiation over North Greenland. However, the expansion of the bare ice area has allowed for more absorption of radiation. With respect to the reference period, our results suggest that there have been regional changes in the seasonal impact of key atmospheric circulation patterns on the SEB components. The impact of climate change was found irrespective of the NAG phase examined, which points to an anthropogenic signal beyond the internal climate variability.

*Data availability.* RACMO2.3p2 (Noël et al., 2019) is available upon contact with Dr. Brice Noël.

*Author contributions.* TS conceptualized the study; BN and WJB provided the model data; SS acquired the model data; TS analyzed the
data; TS, JA, SS, BN, WJB and WS interpreted the results; TS wrote the manuscript with the support of all co-authors.

*Competing interests.* The authors declare that they have no conflict of interest.

*Acknowledgements.* The University of Graz is acknowledged for support of publication costs. The authors thank editor Xavier Fettweis and
the three anonymous referees for the constructive comments.

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
