# Peer review of "The impact of climate oscillations on the surface energy budget over the Greenland Ice Sheet in a changing climate"

_The Cryosphere, 2021_

## Author Comment (AC1)

We would like to thank the editor and the three anonymous referees for their thorough evaluations with constructive comments that certainly will improve the manuscript. In the following, we will address the referees' comments point by point. We mark red the comments given by the referee, give our answers and comments in black and indicate how we addressed the amendments in the manuscript in green.

**Comment on tc-2021-388**
Anonymous Referee #1
Referee comment on "The impact of climate oscillations on the surface energy budget over the Greenland Ice Sheet in a changing climate" by Tiago Silva et al., The Cryosphere Discuss., https://doi.org/10.5194/tc-2021-388-RC1, 2022

*Silva et al. examines the influence of the North Atlantic Oscillation (NAO), Greenland Blocking Index (GBI), and a cluster-aggregation of the aforementioned indices along with integrated water vapor (IWV) over the Greenland Ice Sheet (GrIS) on regional surface energy budget (SEB) changes, derived from the polar-adapted Regional Atmospheric Climate Model (i.e., RACMO2), between the 1959-1990 and 1991-2020 periods. In addition to deconstructing the GrIS-wide and regional SEB and thermodynamic variables (e.g., skin temperature, IWV, and near-surface specific humidity) by phase of these raw and clustered climate indices, the authors also correlate the accumulation and ablation zone rates of change associated with the indices' phases to each of these variables individually for winter and summer seasons. Mesoscale processes forcing SEB changes (e.g., loss of local sea ice increases in wind speeds due to strengthening surface pressure gradients) are also discussed in the context of results. Main conclusions include that GrIS surface warming is most pronounced during winter (following the strongest period of Arctic Amplification), but the associated magnitude and spatial pattern of temperature changes depend on the prevailing atmospheric pattern and presumably its frequency. Meanwhile, sensible heat flux increases in the summer ablation zone are found regardless of the atmospheric circulation pattern, further signaling the importance of mesoscale controls (e.g. katabatic wind strengthening due to land-sea temperature and pressure changes) on low-elevation melt. The level of detail provided in linking the climate indices/oscillations to the SEB and thermodynamic variables is commendable and presents a more detailed picture of atmosphere-GrIS surface forcing than is typically presented in comparable studies.*

*However, this amount of detail also presents challenges with regards to clearly distilling key results. As such, the main findings could be more clearly stated in the abstract and especially in the conclusions. If key takeaways and related points could be more clearly stated through the manuscript, this study could be a valuable addition to the literature.*

*My comments are provided by line number (LN) or specific figure below. While most are minor in nature, the total number of comments may tilt the paper toward the category of major revision.*

We thank Referee #1 for the overall positive assessment. We acknowledge the main concern that also resonates in other reviewers' comments: the fact that we need to state our main findings more clearly. We will integrate this in a revised version as suggested to condense our findings into clear take-home messages. In this way we should be able to address the main criticism appropriately. Below, we list and comment issues of minor nature from Reviewer #1 point by point.

LN14: Do you mean clouds have become optically thinner? Please clarify.
We state that the atmospheric optical thickness has decreased in the southern parts of Greenland. Aside from the cloud's physical state, we do not explore to which extent the decreasing optical thickness is related to other changes in cloud's microphysics properties. What we can add, though, is due to the decrease in water vapor abundance, cloud's formation becomes improbable in the region.

LN43: Can you clarify what is intended and ultimately hypothesized by "tilt within largescale structures may have an impact at different locations"? Work by Woollings et al. (2008) J. Climate and Hanna et al. (2018) Int. J. Climatol. has shown that that the setup of Greenland blocks tends to precede by a couple of days downstream positive North Atlantic SLP anomalies in the vicinity of the Icelandic Low (i.e., -NAO conditions). Perhaps referencing this work may help clarify the large-scale structural reference?
The references suggested are indeed relevant for the description of the large-scale systems in the North Atlantic and we will include them once we submit a revised version. We will also reformulate the paragraph addressing the hypothesis mentioned by the Referee. The vertical tilt (of temperature and geopotential) within large-scale systems exists due to baroclinicity and recently has been pointed out by Martineau et al. (2020) as an essential mechanism in the North Atlantic for large-scale system development. Therefore, we hypothesize that the tilt within large-scale structures plays a role when calculating climate oscillations, which rely on one parameter at one specific atmospheric

level (only at 500 hPa or only at the surface). We thus suggest various spatiotemporal effects on the near-surface impacts dependent on the climate oscillation in use. Particularly in the cold season and under strong cyclonic influence, the usage of a classification that combines NAO and GBI rather than select one of these climate oscillations in isolation may help to account for processes in different atmospheric levels as the air mass properties are distinct (e.g., frontal systems).

In fact, the seasonal composites made for each NAG (Fig. S4) show that the location of the maxima of geopotential height at 925 and 500 hPa, respectively, are typically hundreds of kilometers apart. These distances are likely to be larger for analysis in higher temporal resolutions.

[Figure]

**Figure S4.** Inter-seasonal 500 hPa geopotential height anomaly (contour lines; positive: solid and negative:dashed; spaced in 10 m intervals), and 925 hPa geopotential height anomaly for each NAG phase with respect to climatology (1959-2000). The ridge (trough) at 925 and 500-hPa geopotential height anomaly is indicated as dot and cross, respectively, for +NAG(−NAG).

LN76: It would be good to emphasize around this point in the introduction the explicit goal and primary research questions of the study. These would help build upon some previously mentioned hypotheses (e.g., LN 74-75) by adding more structure and thus guidance for the reader toward analyses that lie ahead.

Thanks, we changed this as suggested. We initially had decided to state each of our hypotheses next to the respective literature review paragraph (1st hypothesis in LN 43 and 2nd hypothesis in LN 74) instead of a paragraph explicitly dedicated to our research goals. We will follow the referee's suggestion to emphasize at the beginning of the last introductory paragraph our research aims.

Here, we explore a cluster method that links the role of the North Atlantic Oscillation (NAO) with the prevailing mid-troposphere circulation pattern over Greenland, commonly known as Greenland Blocking Index (GBI), along with the atmospheric water vapor over the GrIS. Additionally, we investigate the regional impact on the surface energy budget (SEB) of contrasting atmospheric circulation clusters, and finally, we examine changes on SEB components within clusters by comparing recent decades (1991-2020) to a historical period (1959-1990).

…

Section 2 describes the data analyzed, explains the clustering method, and justifies 1991 as the period breakpoint. The Results and Discussion is broken into three subsections. In Section 3.1 we show the inter-annual variability of the newly cluster-derived classification and compare it with NAO and GBI alone; in Section 3.2 we describe the inter-seasonal and regional variability of the cluster classification; in Section 3.3 we present spatio-temporal anomalies within the same atmospheric circulation cluster, and finally, we focus our discussion on the regional changes in the summer ablation zone.

LN 117: List the flux terms units, W/m^2?

Thanks, we include the flux terms units as suggested.

LN126-127: I think you could move this sentence (beginning with "Using the 62 years…") to L131 and explicitly list the sub-periods that were the result of equally dividing the total years in the dataset.

Thanks, we took into consideration the Referee's input! Since Referee #3, recommended to make this particular section clearer in the manuscript, we reformulated the paragraph. This improved paragraph contains an explanation of trends in sub-periods, a definition of the trend ratio, an example of one period ratio and most importantly we explicitly state the reason for the breakpoint detection.

Most studies agree that the pronounced Greenland summer mass loss started in the 1990s (e.g. Mouginot et al. 2019, Hanna et al. 2021, Shepherd et al. 2020). However, the onset of a clear negative trend varies depending on the time period of each study and on the dataset used. In order to determine the breakpoint of the marked summer mass loss in RACMO2, we divided the GrIS into its main seven drainage basins (see Fig. S1) and run regional trends for multiple periods with multiple sizes for the 62 years of data. This will allow the investigation of atmospheric and glaciological conditions prior and post the breakpoint. The breakpoint was formed on the most regionally frequent and the largest absolute trend ratios. One trend ratio (RT) is based upon two slopes from equally-sized sub-periods that are split in a common central year. The division of the slope after the central year (s2) by the slope before the central year (s1) produces one RT. For instance, s1 between 1977 and 1995 and s2 between 1995 and 2013, whose central year is 1995 (Fig. 1) gives RT > 1. This means that s2 is more pronounced than s1. The length of each sub-period varied from 15 to 32 years.

The non-parametric Mann-Kendall (M-K) trend test is used to assess trend monotonicity and significance on summer surface ablation rates (c.f. Section 2.2). The slope corresponds to the Theil-Sen (T-S) estimator. The T-S estimator is a robust regression method that does not require the data to be normally distributed and is hence less vulnerable to outliers. One specific period is considered significant only when the confidence level from the M-K test is higher than (or equal to) 90% in both sub-periods. Trends in periods exhibiting confidence levels lower than 90% may still be identical to those exhibiting greater significance levels but given their high variability they were not considered.

As we analyzed 612 trends for equally-sized sub-periods, we consider it unfeasible to explicitly list them. Nevertheless, we show to the Referee in the figure below the distribution of sub-periods used.

[Figure]

Figure R1-Distribution of sub-periods used for the trend analysis in Section 2.3

L155: Both the NAO and GBI indices should be defined here (e.g., domains, methods, papers defining such, etc). Moreover, both atmospheric indices are derived at z500, did you look at the surface NAO (i.e., Hurrell PCA or weather station-based NAO)? In this context, I recommend in the paper that you address why only z500 indices are used or why NAO from SLP data is not used. This discussion would seem appropriate since you are exploring through cluster analysis how co-varying characteristics of these atmospheric patterns (along with IWV) may impact GrIS surface conditions.

[Figure]

Thanks, this indeed is a crucial remark to highlight in the methods and discussion, as the NAO can be calculated with different methods (e.g., weather stations, PCA and clustering) at different pressure levels (e.g. surface and 500 hPa).

We follow up on this by statistically comparing two datasets using PCA: the surface (Hurrel) NAO (from NCAR) and the 500 hPa NAO (from NCEP/CPC). Relative differences on NAO percentile and NAG classifications dependent on the NAO used are shown in Figure 2. The NAO as derived from different levels is highly and positively correlated. However, the NAG clustering and the NAO percentile classification are sensitive to the dataset used. Relative differences in the NAG classification are lower than 10% and are essentially related to the separation with the neutral phase. Typical NAO classifications using the 1st and 3rd quartile to respectively categorize negative and positive phases show (up to 10 times) larger disagreements than the NAG

Figure R2- Relative differences on NAO percentile (gray) and NAG (black) classifications dependent on the NAO used. The first (last) three bars correspond to winter (summer) phases.

classification. The disagreement between the NAO from different levels roses from variations at the 500 hPa geopotential height anomalies that are not resembling with surface anomalies.

The NAG results from Figures 4 and 5 are dependent little on the NAO definition used as all spatial anomalies remain. In general, the spatial anomalies using surface NAO are larger and more significant than for 500hPa NAO. However, spatial anomalies based on the percentile classification depend greatly on the pressure level used for deriving NAO, as the spatial anomalies for surface and 500hPa are different (occasionally, with opposite signals) in some cases.

Once again, we appreciate the referee's remark that will make our study more comparable and applicable. We will hence adapt our results using the definition of NAO by Hurrel (PCA based). In addition, we will briefly highlight differences between both NAO methods.

LN167-168: To clarify the sentence, I recommend substituting "predominant" with "prevailing" then remove "prevailing" in LN168.

Thanks, we changed this as suggested.

LN172: The clustering approach could use more description. Why did you pick 3 clusters? Did you select these based on subjective or objective criteria? Are the results sensitive to the number of clusters selected and analyzed?

The selection of 3 clusters (negative, neutral, and positive) is based on the number of climate oscillation phases commonly used in many studies (e.g. Croci-Maspoli et al. (2007), Gimeno, et al. (2002), Wu and Zhang (2015)). We believe that the inclusion of more clusters is not useful, as they become abstract and objectively challenging to physically interpret them. As shown above, the results are sensitive to the selected variables (NAO, GBI, GrIS IWV) in the cluster, as well as the number of clusters set.

LN179: Be more specific on what data is shown in the Figure S3 scatterplots. This is very vague as currently written.

Thanks, we will improve Figure S3 caption to:

Seasonal NAG phases based on the k-means clustering: positive (red cluster); neutral (black cluster); and negative (blue cluster) phase.

NAO and GBI phases are categorized based on the seasonal 25th (blue line) and 75th (red line) percentile. The Spearman correlation coefficient (rS) is shown for clusters with significance higher than 90%.

The third-dimension correspondent to GrIS IWV is omitted. The dark circle indicates years whose seasonal GrIS IWV is greater than the 95th percentile.

LN183: "The positive phase of NAG is connected…" Is this "connection" illustrated somewhere either graphically or statistically? This would be helpful to show the reader to see what +NAG entails.

We show the NAG and its relations in Figure 2, S3 and S4:

Figure 2 illustrates the development of NAO and GBI over time. Figure S3 shows the relationship between NAO and GBI and the respective NAG clusters/phases. Figure S4 characterizes the anomalies of geopotential height at 500 hPa with respect to the climatology dependent on season and NAG phase.

We complete the sentence with: Based on resulting clustering (Fig. S3) and the involved large-scale structures (Fig. S4), the positive phase of NAG is connected to the …

Figure 2 caption: The last sentence is unclear. I suggest mentioning that data from 1991 onward is found to the right of the gray vertical line.

We expand this and add:

For reference, 1991 is highlighted to illustrate GBI, NAO and NAG phases, as well as seasonal accumulated SMB in the time-series.

L192: Why show the 925 hPa height anomaly rather than SLP, a field typically used in defining surface pressure characteristics of the NAO.

For practical reasons, model pressure levels were preferred, but for clarity we can easily replace 925 hPa by surface pressure in a future version.

L193: "vertical tilting structure" meaning what? Please clarify.

We expand this definition stating:

Under strong baroclinicity, the minimum/maximum in geopotential (Fig. S4) and temperature are vertically titled within large-scale structures.

L199: Do you mean "winter" instead of "spring" is when the equator-to-pole air temperature contrast is maximized?

On average, the strongest equator-to-pole temperature gradient coincides with the end of winter. Depending on the year, the maximum equator-to-pole air temperature contrast takes place between February and March, and therefore coincides either in the meteorological late winter or early spring. In order to avoid misinterpretations, we drop the clause ("when the equator-to-pole temperature is the strongest") out of the sentence.

Figure 3: Should the colorbar label r_s(Seasonal ablation…) be r^2 as it is in the caption?

While the figure colors are representative of the Spearman correlation ($r_S$) and vary from -1 to 1, the noted number ($r_S^2$) indicates the proportion of shared variance among ranked variables and ranges between 0 and 1. We reformulate the second sentence of the figure caption to: The Spearman correlation coefficient ($r_S$) is color-coded and the determination coefficient ($r_S^2$) is displayed.

Figure 4: "The percentage of each NAG phase used..." can you please clarify what this means? As I interpret it, it sounds like some +/-NAG days were composited and some were not without explanation as to why since f0 and f-percentages do not sum to 100% (as in Fig 5).

Thanks for pointing this out. This was indeed misleading. The relative frequency of seasonal NAG (0NAG, +NAG, -NAG) sums to 100%. However, as the 0NAG is the reference for +NAG and -NAO, we indicated its frequency at both instances. We improve the mentioned caption sentence to: The seasonal frequency (f in %) of each NAG phase (0NAG, +NAG, -NAG) used to produce composites is indicated as subtitle.

LN253: Remove "configuration."

Thanks, we removed this as suggested.

Also, since these surface temperature and radiative fields (i.e., Fig 5) increase regardless of atmospheric pattern, does that suggest that warming climate is the main culprit in driving these fluxes that impact SMB? What link is being made with adjacent marginal seas; they respond similarly to these fields as the GrIS?

Our results suggest warming climate and its spatial magnitude over the GrIS depending on the NAG phase. As we supposedly split internal climate variability into clusters, we point anthropogenic effects as the culprit in the Conclusions. However, a direct attribution to individual factors of such a complex system is indeed very challenging and beyond our study scope.

Temporal changes over the adjacent seas depending on season and NAG phase are described and compared to GrIS along the Results and Discussion (e.g., LN261, LN279 and LN305). However, we acknowledge that this point is a missing point in the Conclusions, and therefore, we will add it in a revised version.

LN 270-271: In comparing 1991-2020 against the reference period, do you mean "increased surface-based inversions…"

We simply adopt the hypothesis from other studies (e.g., Niwano et al. 2019) stating that surface-based inversions in combination with subsidence are the physical mechanism responsible of the elevated values of water vapor near the surface. To avoid confusion, we reformulate the sentence to: As a consequence of the surface-based inversions favored by subsidence and surface melt, the surface comprises more q2m for the period 1991-2020 than during the reference period.

LN272-273: This sentence is confusing; the IWV increase over the northern GrIS is not related to local cloud water content? Please clarify. Is this shown in the analysis and if so, then where?

The IWV increase is not only related to the local cloud water content changes, but also a result of the increasing near-surface water vapor. We do not attribute the increase in SWnet to cloud water content changes in the Northern regions because the SWin and cloud water content did not significantly change in comparison to the reference period (Fig. S12). We improve the sentence to: Despite the residual change in SWin (Fig. S12) and cloud water content, the SWnet (Fig. S14) has significantly increased over the northern regions. Thus, we attribute the increase in IWV to the widespread increased q2m.

LN293: "high summer GBI values…" –where is this analysis shown?

In order to reduce the number of supplementary figures, we removed seasonal GBI and NAO time-series. However, this statement is supported in Figure 2 as seen by the red shading enhancement and frequency of positive GBI greater than the third quartile (marked by diamonds). In addition, we will also add a few studies entirely focused on the recent extreme GBI (e.g., Barrett et al. 2020) and the Greenland Blocking (e.g., Wachowicz et al 2020).

LN 297: "crucial role of NAO advecting heat and moisture…" through storms/the storm track migrating poleward toward Greenland?

We complete the sentence taking into consideration the reviewer's remark.

LN311-313: This sentence is a bit hard to follow. I recommend splitting it into two sentences.

We follow the reviewer's advice and change this paragraph to: The near-surface temperature gradient is enhanced by the larger increase of the air temperature than the temperature of the melting snow/ice surface that is physically limited to 273.15 K. The melting snow/ice surface, in conjunction with steep slopes, promotes downslope winds. In addition, the migration of the snowline to higher elevations of the ice sheet enhances the surface pressure gradient and adds momentum to the flow contributing to the observed wind speed strengthening.

LN315: To clarify, is the suggestion that the summer wind speed increase over northern GrIS is due to the near-complete summer melt of Baffin (particularly) ice cover, a typical feature of its annual cycle? Atmospheric circulation patterns can accelerate the melt, but their intensity and orientation could also presumably affect the onset of such summer wind increases.

We agree with the reviewer: atmospheric circulation patterns can influence the onset of the regional sea ice melt. We will add the Referee's thought as follows: The wind speed increase in the Northwest ablation zone is only significantly different from the reference period under 0NAG (Fig. S12 and S13). The seasonality of the almost sea-ice free Baffin Bay (Bi et al. 2019) can be one of the factors that impedes statistical evidence in wind speed increase in the region. This is in contrast with the North and Northeast zones, where coastal margins are almost permanently ice covered during the reference period and the wind speed has increased significantly regardless of the NAG phase. The increased wind speed may also form brash ice which promotes more convection, and consequently, generates a low surface pressure in the neighboring seas that will enhance the regional surface pressure gradient leading to higher wind speeds.

LN 322-324: It would be a good idea to direct the reader to this figure or analysis within the paper.

We follow the advice and add the figure (Fig. 6) in the main paper in the revised version.

LN382: Change "has" to "have"

Thanks, we changed this as suggested.

**Supplemental Material:**

Figure S8: Label seasons at the top of the graphic as with Figure S7, etc.

Thanks, we labeled this as suggested.

Figures S12-14: I am confused what these graphics actually show and what the units on each concentric circle represent. Please clarify.

Thanks, we expand their description in their caption stating:

Fig. S12. Changes in atmospheric variables contributing to SEB components between 1959-1990 and 1991-2020 in the summer ablation zone for each NAG phase (color-coded). Variables in the panel a) exhibit the variable name and the respective unit. All variables in the panel b) are fluxes in W m$^{-2}$. The spatial relative frequency of ablation is shown at the center. Negative changes are limited by the gray area. Hollow circles indicate significant mean differences based on the Wilcoxon rank-sum statistic test for unpaired sets with a confidence higher than 90%.

These charts come to support the analysis in Figure 6. The radar chart aspect is an alternative to the bar chart in Figure 6.

**References:**

- Barrett, B. S., Henderson, G. R., McDonnell, E., Henry, M., and Mote, T.: Extreme Greenland blocking and high-latitude moisture transport, Atmospheric Science Letters, 21, e1002, https://doi.org/10.1002/asl.1002, 2020.
- Bi, Haibo, et al. "Baffin Bay sea ice inflow and outflow: 1978–1979 to 2016–2017." The Cryosphere 13.3 (2019): 1025-1042.
- Croci-Maspoli, Mischa, Cornelia Schwierz, and Huw C. Davies. "Atmospheric blocking: Space-time links to the NAO and PNA." Climate Dynamics 29.7 (2007): 713-725.
- Gimeno, Luis, et al. "Identification of empirical relationships between indices of ENSO and NAO and agricultural yields in Spain." Climate research 21.2 (2002): 165-172.
- Niwano, Masashi, Akihiro Hashimoto, and Teruo Aoki. "Cloud-driven modulations of Greenland ice sheet surface melt." Scientific reports 9.1 (2019): 1-8.
- Wu, Zhiwei, and Peng Zhang. "Interdecadal variability of the mega-ENSO–NAO synchronization in winter." Climate dynamics 45.3 (2015): 1117-1128.
- Wachowicz, L. J., Preece, J. R., Mote, T. L., Barrett, B. S., and Henderson, G. R.: Historical Trends of Seasonal Greenland Blocking Under Different Blocking Metrics, Int J Climatol, 41, E3263–E3278, https://doi.org/10.1002/joc.6923, 2020.

---

## Author Comment (AC3)

We would like to thank the editor and the three anonymous referees for their thorough evaluations with constructive comments that certainly will improve the manuscript. In the following, we will address the referees' comments point by point. We mark red the comments given by the referee, give our answers and comments in black and indicate how we addressed the amendments in the manuscript in green.

**Comment on tc-2021-388**
Anonymous Referee #3
Referee comment on "The impact of climate oscillations on the surface energy budget over the Greenland Ice Sheet in a changing climate" by Tiago Silva et al., The Cryosphere Discuss., https://doi.org/10.5194/tc-2021-388-RC3, 2022

**General comments**
*In this study, the authors use a cluster analysis of NAO, GBI, and column water vapor to derive a "North Atlantic influence on Greenland" (NAG) index. RACMO2 output is then used to investigate atmospheric and cryospheric conditions across different NAG phases and their changes across a 1991 break point in summer surface mass loss. Results describe a large array of seasonal anomalies in atmospheric conditions and surface energy balance components across the NAG phases for each season during the pre- and post-1991 periods.*
*I found this paper difficult to follow due to the large number of figures and sub-panels within figures and the organization of the paper, as it lacks a clear statement of the research questions or summary of what important new information was learned in this study. I also agree with the editor that there is insufficient originality, at least with how the results are presented in current form. However, there do not appear to be any technical flaws in the methods employed, and I do think there is potential for some of the results to form a nice study if they are better organized. I encourage the authors to think about what they consider to be the most important and novel findings contained within their many analyses and distill these findings into a focused message for readers to take from the paper. As an example, the contrast between moistening in northern Greenland and drying / clearing in southern Greenland under +NAG conditions is an interesting finding. In addition to the specific comments and technical corrections below, I would recommend that the authors simplify the figures, and restructure the discussion so that a large part of the findings in the main paper are not describing figures found in the supplement.*

We thank Referee #3 for the challenging remarks and appreciate the overall potential they see in our study. We acknowledge that there was indeed an obviously too unclear aim of the study which was similarly pointed out by Referee #1 and we will implement this very thoroughly in the revised version. We also take up the reviewer's recommendation and will strengthen the organization of the discussion in the revised version. This will significantly improve the reading of the paper. For now, we would point to the amendments made and the following major research questions that we will state at the end of the revised introduction:

Here, we explore a cluster method that links the role of the North Atlantic Oscillation (NAO) with the prevailing mid-troposphere circulation pattern over Greenland, commonly known as Greenland Blocking Index (GBI), along with the atmospheric water vapor over the GrIS. Additionally, we investigate the regional impact on the surface energy budget (SEB) of contrasting atmospheric circulation clusters, and finally, we examine changes on SEB components within clusters by comparing recent decades (1991-2020) to a historical period (1959-1990).
…
Section 2 describes the data analyzed, explains the clustering method, and justifies 1991 as the period breakpoint. The Results and Discussion is broken into three subsections. In Section 3.1 we show the inter-annual variability of the newly cluster-derived classification and compare it with NAO and GBI alone; in Section 3.2 we describe the inter-seasonal and regional variability of the cluster classification; in Section 3.3 we present spatio-temporal anomalies within the same atmospheric circulation cluster, and finally, we focus our discussion on the regional changes in the Furthermore, we will simplify the figures whenever appropriate.

**Specific comments**
Did the authors examine trends in the frequency of NAG phases, or did they only look at changes in atmospheric conditions over time during each NAG phase?

[Figure]

[Figure]

We made a brief trend analysis in the frequency of the NAG phases. We collected decadal-moving proportions as dependent on the NAG phase (Figure R1). However, we solely focused on spatio-temporal changes of the atmospheric conditions within NAG phases overtime. The NAG development over time is found in Figure 2.

L18–45: The opening paragraph of the Introduction is quite long and does not provide a compelling introduction to the research topic that the authors investigate. I think it would make more sense to first introduce the problem of Greenland surface melt and its atmospheric drivers (the second paragraph), before moving on to the indices that are used to help quantify these atmospheric drivers (first paragraph).

We acknowledge the referee's comment, which was also mentioned by Referee #1. We will reformulate the Introduction by moving the first paragraph and breaking it into two parts in the revised version.

L21: Liu and Barnes (2015) is a good reference on the relationship between Rossby wave breaking and poleward moisture transport in the vicinity of Greenland.

Thanks for pointing us towards this reference which we will implement both in the introduction and in the discussion in a revised version.

L29: I'm not sure it's correct to say that the NAO phase "explains most of the heat and moisture transported poleward". It's more accurate to say that the NAO phase affects the location and magnitude of poleward heat and moisture transport, and provide a reference on this.

Indeed, the formulation was misleading, and we adopt the formulation suggested by the referee. At the same time, we add the following references (e.g., Bjørk et al., 2018, Papritz et al, 2020).

L32: GBI simply quantifies the mean 500 hPa geopotential height over a Greenland centered domain, as the authors state in the previous sentence. It does not directly quantify the strength and moisture transported over the Greenland domain although it is correlated with these quantities (see the Barrett et al. 2020 paper the authors already cite). See Wachowicz et al. 2020 for a more nuanced discussion of the GBI and comparison with other blocking metrics.

To our understanding, GBI provides insight on blocking-like conditions in the vicinity of Greenland. We agree that "quantify the strength" was not the best wording. Nevertheless, positive GBI conditions can regionally block heat and moisture transport towards the interior of the GrIS. We reformulate the sentence as: Its index denotes the predominant atmospheric circulation pattern in the vicinity of Greenland, and it regionally controls the heat and moisture transported towards the interior of the GrIS.

L120 and Figs. 5, S9–S11: It is not clear how the method of dividing the adjacent seas into four areas is actually used to assess potential sources of moisture. I am having trouble understanding what the numbers in the corners of Figs. 5 and S9–S11 (the "differences in composites between adjacent seas") represent.

We were interested to know if changes in atmospheric variables over land were similar to the changes over the adjacent seas, as moisture and heat are advected from there towards the ice sheet. In order to simplify the visualization, instead of showing the spatial anomalies over the adjacent seas we preferred to show the temporal change in one value, which is displayed in the corners of each subpanel. We acknowledge the complexity of the figure, but at the same time we value its contribution to the discussion. We will move the 0NAG to supplementary and improve the caption description to: Seasonal and spatial anomalies for (a) integrated water vapor, (b) incoming longwave radiation reaching the surface, (c) near-surface specific humidity, and (d) skin temperature between 1959-1990 and 1991-2020 as dependent

on the NAG phase from RACMO2.3.p2. The percentage (f) of the NAG phase in each period is indicated for each season. For reference, Summit and South Dome are marked as big and small triangles, respectively. Stippled regions indicate areas with a confidence level greater than 90% (based on the Wilcoxon rank-sum statistic test for unpaired sets). Temporal anomalies between composites over the adjacent seas are also shown as colored numbers (Baffin Bay: upper left; Greenland Sea: upper right; Irminger Sea (lower right) and Labrador Sea (lower left). See Figure S1 to discern the extension overseas and, Figure Sx to examine spatio-temporal anomalies under 0NAG.

L123: It should be stated explicitly at the beginning of section 2.3 that the reason for the break point detection is to form the basis for subsequent analyses of atmospheric and glaciological conditions before and after the break point. As it stands now, this section reads like it is reporting research findings, rather than describing a method that will be used to produce the results of the study.

We fully agree with the comment. Also, taking into consideration the comment from Referee #1, we reformulated Section 2.3 to:

Most studies agree that the pronounced Greenland summer mass loss started in the 1990s (e.g. Mouginot et al. 2019, Hanna et al. 2021, Shepherd et al. 2020). However, the onset of a clear negative trend varies depending on the time period of each study and on the dataset used. In order to determine the breakpoint of the marked summer mass loss in RACMO2, we divided the GrIS into its main seven drainage basins (see Fig. S1) and run regional trends for multiple periods with multiple sizes for the 62 years of data. This will allow the investigation of atmospheric and glaciological conditions prior and post the breakpoint. The breakpoint was formed on the most regionally frequent and the largest absolute trend ratios. One trend ratio (RT) is based upon two slopes from equally-sized sub-periods that are split in a common central year. The division of the slope after the central year (s2) by the slope before the central year (s1) produces one RT. For instance, s1 between 1977 and 1995 and s2 between 1995 and 2013, whose central year is 1995 (Fig. 1) gives RT > 1. This means that s2 is more pronounced than s1. The length of each sub-period varied from 15 to 32 years.

The non-parametric Mann-Kendall (M-K) trend test is used to assess trend monotonicity and significance on summer surface ablation rates (c.f. Section 2.2). The slope corresponds to the Theil-Sen (T-S) estimator. The T-S estimator is a robust regression method that does not require the data to be normally distributed and is hence less vulnerable to outliers. One specific period is considered significant only when the confidence level from the M-K test is higher than (or equal to) 90% in both sub-periods. Trends in periods exhibiting confidence levels lower than 90% may still be identical to those exhibiting greater significance levels but given their high variability they were not considered.

L172–180: State up front that you are using a k-means clustering method (rather than first describing the method and naming it as k-means clustering at the end of the description).

Thanks, we will name the clustering method before its description.

L181: I don't think the "influence of the North Atlantic over Greenland" is an accurate description of what the NAG index produced by the cluster classification provides. Maybe describe as the "influence of regional climate" on Greenland instead. (The AMV index, which specifically quantifies oceanic conditions, is discussed in the Introduction and in L159 in the Data and Methods, but doesn't appear to be included as an input to the NAG index.)

We name NAG as such given the importance of the North Atlantic Oscillation in setting the storm track and atmosphere-ocean interactions feeding large-scale weather systems along the North Atlantic. As mentioned in the Introduction, the AMV index was negative until the early 1990's and it has been positive since. Given the strong relationship in summer between AMV and GBI (and GrIS IWV), the inclusion of AMV alongside NAO, GBI and GrIS IWV in the 62 years of cluster analysis does not contribute with new information and it does not change the summer classification. However, in the remaining seasons (e.g. spring), positively high AMV in recent years adds noise to the classification by making GBI<0 and NAO>0 as +NAG, which is contradicting with the rest of the cluster.

L226, 232–236, 311–315, and 368–370: The authors should consider that the stronger wind speeds during the +NAG phase are not strictly katabatic but are enhanced by the interaction of a strengthened synoptic-scale pressure gradient with the Greenland ice sheet's orography. I would suspect this is especially true for the winter cases where the authors find that increased wind speeds and SHF occur during +NAG. Previous studies have described this synoptically-driven wind enhancement as the Greenland "barrier jet" or "plateau jet" – see e.g. Meesters 1994, van den Broeke and Gallée 1996, Moore et al. 2013, Mattingly et al. 2020.

We will improve our discussion regarding the coupling of the katabatic winds with upper air winds whenever appropriate.

L255, Figs. 4–6: I assume all the results in Figs. 4–6 (e.g. the increasing trend in TCWV in northern Greenland described in L255) are produced from RACMO2 data? If so this should be explicitly stated in the figure captions and the text.

We add to the figure caption that these results are based on the RACMO2.3p2 output, and convenient figure reference along the text.

L282–284: This statement about the seasonal preconditioning effect of skin temperature warming appears to contract the finding in L166–168 that there is no relevant time-lag response between seasonal GrIS surface mass fluxes and the predominant atmospheric circulation pattern prevailing in the preceding seasons.

The cross-correlation in L166–168 was applied to the overall GrIS SMB with isolated climate oscillations. In L282–284, we discuss that the skin temperature in winter during the period 1991-2020 is warmer than in 1959-1990 independently of the NAG phase. Hence, we do not attribute the impact of changing Tskin during individual subsequent seasons, which is why a direct comparison among the statements is not possible. We could add a clarification as "Note, that Fig. 5 compares general conditions among different periods and cannot be used in order to deduce subsequent season's cause and effect". If the Referee does not find this answer satisfactory, we kindly ask you to reformulate how can these statements be contradicting.

L314: How would decreasing ice in neighboring seas contribute to an increase in summer wind speed? Please explain in more detail.

A direct attribution to individual components of such a complex system is indeed very challenging. However, in general, more open waters promote more convection, and consequently, generate a low surface pressure in the neighboring seas that will enhance the regional surface pressure gradient leading to higher wind speeds. Detailed modelling studies may shed light on such relevant connections. They do, however, go beyond the scope of the current study. We will address this point in a revised version discussing the potential of testing such hypotheses with an RCM on regional to local scales.

**Technical corrections**

L2: The word "fluxes" is not needed since "advection" already describes the horizontal flow of heat and moisture.

Thanks, we changed this as suggested.

L2: surface mass balance of what? (state definitively that it's the SMB of the Greenland Ice Sheet)

Thanks, we changed this as suggested.

L2: "pattern" --> "patterns"

Thanks, we changed this as suggested.

L14: "optical" --> "optically"

Thanks, we changed this as suggested.

L14–16: Run-on sentence. Consider splitting into two sentences.

We follow the advice and split up the sentence in two:

In the southern part of Greenland, the atmosphere has gotten optically thinner, thus allowing more incoming shortwave radiation to reach the surface. In the northern part, the incoming shortwave radiation has changed little with respect to the reference period, but the surface albedo decreased due to the expansion of the bare ice area.

L15: "shortwave radiation flux" should be "shortwave radiation" or "shortwave radiative flux"

Thanks, we changed this as suggested.

L18: north of the *climatological location of* the jet stream

Thanks, we clarify this ambiguity by changing the sentence to:

The GrIS is most commonly found north of the jet stream

L63: "largest" --> "most intense"?

Thanks, we follow this advice and reword accordingly

L91: ERA5 is the most recent reanalysis product from ECMWF (it's not an "earlier product")

Thanks, we agree with the Referee and reword to: The ECMWF reanalyses products - ERA40 (Uppala et al., 2005) (1959-1978); ERA-I (1979-1989); and ERA5 (1990-2020) - are used to laterally force…

L211: The abbreviation "0NAG" is used repeatedly from this point forward without previously being defined in the text. It appears to be defined in the caption for Figure 4, but its meaning should be explicitly stated in the text at first use.

Thanks, we add its meaning to the beginning of Section 3.2: Spatial and inter-seasonal anomalies under contrasting NAG (+/-) phases with respect to the neutral phase (0NAG) are illustrated in Figure 4…

L331: Delete the word "or" at the end of this line.

Thanks, we deleted "of" at the end of this line.

L335: Accumulation zone has been decreasing *in area*?

Yes, in area, we add this at the respective line in order to clarify.

L337: Insert the word "zone" after "accumulation"

Thanks, we changed this as suggested.

L366-367: "vertically distributed changes" --> "vertical distribution of changes"

Thanks, we changed this as suggested.

**References**

Liu, C. and Barnes, E. A.: Extreme moisture transport into the Arctic linked to Rossby wave breaking, J. Geophys. Res. Atmos., 120, 3774–3788, https://doi.org/10.1002/2014JD022796, 2015.

Mattingly, K. S., Mote, T. L., Fettweis, X., van As, D., Van Tricht, K., Lhermitte, S., Pettersen, C., and Fausto, R. S.: Strong Summer Atmospheric Trigger Greenland Ice Sheet Melt through Spatially Varying Surface Energy Balance and Cloud Regimes, J. Climate, 33, 6809–6832, https://doi.org/10.1175/JCLI-D-19-0835.1, 2020.

Meesters, A.: Dependence of the energy balance of the Greenland ice sheet on climate change: Influence of katabatic wind and tundra, Q.J Royal Met. Soc., 120, 491–517, https://doi.org/10.1002/qj.49712051702, 1994.

Moore, G. W. K., Renfrew, I. A., and Cassano, J. J.: Greenland plateau jets, Tellus A: Dynamic Meteorology and Oceanography, 65, 17468, https://doi.org/10.3402/tellusa.v65i0.17468, 2013.

van den Broeke, M. R. and Gallée, H.: Observation and simulation of barrier winds at the western margin of the Greenland ice sheet, Q.J Royal Met. Soc., 122, 1365–1383, https://doi.org/10.1002/qj.49712253407, 1996.

Wachowicz, L. J., Preece, J. R., Mote, T. L., Barrett, B. S., and Henderson, G. R.: Historical Trends of Seasonal Greenland Blocking Under Different Blocking Metrics, Int J Climatol, 41, E3263–E3278, https://doi.org/10.1002/joc.6923, 2020.

---

## Author Response (AR1)

We would like to thank the editor and the three anonymous referees for their thorough evaluations with constructive comments that certainly will improve the manuscript. In the following, we will address the referees' comments point by point. We mark red the comments given by the referee, give our answers and comments in black and indicate how we addressed the amendments in the manuscript in green.

**Comment on tc-2021-388**

Anonymous Referee #1

Referee comment on "The impact of climate oscillations on the surface energy budget over the Greenland Ice Sheet in a changing climate" by Tiago Silva et al., The Cryosphere Discuss., https://doi.org/10.5194/tc-2021-388-RC1, 2022

*Silva et al. examines the influence of the North Atlantic Oscillation (NAO), Greenland Blocking Index (GBI), and a cluster-aggregation of the aforementioned indices along with integrated water vapor (IWV) over the Greenland Ice Sheet (GrIS) on regional surface energy budget (SEB) changes, derived from the polar-adapted Regional Atmospheric Climate Model (i.e., RACMO2), between the 1959-1990 and 1991-2020 periods. In addition to deconstructing the GrIS-wide and regional SEB and thermodynamic variables (e.g., skin temperature, IWV, and near-surface specific humidity) by phase of these raw and clustered climate indices, the authors also correlate the accumulation and ablation zone rates of change associated with the indices' phases to each of these variables individually for winter and summer seasons. Mesoscale processes forcing SEB changes (e.g., loss of local sea ice increases in wind speeds due to strengthening surface pressure gradients) are also discussed in the context of results. Main conclusions include that GrIS surface warming is most pronounced during winter (following the strongest period of Arctic Amplification), but the associated magnitude and spatial pattern of temperature changes depend on the prevailing atmospheric pattern and presumably its frequency. Meanwhile, sensible heat flux increases in the summer ablation zone are found regardless of the atmospheric circulation pattern, further signaling the importance of mesoscale controls (e.g. katabatic wind strengthening due to land-sea temperature and pressure changes) on low-elevation melt. The level of detail provided in linking the climate indices/oscillations to the SEB and thermodynamic variables is commendable and presents a more detailed picture of atmosphere-GrIS surface forcing than is typically presented in comparable studies.*

*However, this amount of detail also presents challenges with regards to clearly distilling key results. As such, the main findings could be more clearly stated in the abstract and especially in the conclusions. If key takeaways and related points could be more clearly stated through the manuscript, this study could be a valuable addition to the literature.*

*My comments are provided by line number (LN) or specific figure below. While most are minor in nature, the total number of comments may tilt the paper toward the category of major revision.*

We thank Referee #1 for the overall positive assessment. We acknowledge the main concern that also resonates in other Referees' comments: the fact that we need to state our main findings more clearly. We will integrate this in a revised version as suggested to condense our findings into clear take-home messages. In this way we should be able to address the main criticism appropriately. Below, we list and comment issues of minor nature from Referee #1 point by point.

LN14: Do you mean clouds have become optically thinner? Please clarify.

We state that the atmospheric optical thickness has decreased in the southern parts of Greenland. Aside from the cloud's physical state, we do not explore to which extent the decreasing optical thickness is related to other changes in cloud's microphysics properties. After reformulating the Abstract the equivalent statement commented by the Referee became: In the southern part of Greenland, the atmosphere has gotten optically thinner due to the decrease in water vapor thus allowing more incoming shortwave radiation to reach the surface.

LN43: Can you clarify what is intended and ultimately hypothesized by "tilt within largescale structures may have an impact at different locations"? Work by Woollings et al. (2008) J. Climate and Hanna et al. (2018) Int. J. Climatol. has shown that that the setup of Greenland blocks tends to precede by a couple of days downstream positive North Atlantic SLP anomalies in the vicinity of the Icelandic Low (i.e., -NAO conditions). Perhaps referencing this work may help clarify the large-scale structural reference?

The references suggested are indeed relevant for the description of the large-scale systems in the North Atlantic and we included them in the revised Introduction (e.g. LN 57, LN 209) and Discussion (LN326).

We also reformulated the paragraph addressing the hypothesis mentioned by the Referee in LN74 to: The vertical tilt (of temperature and geopotential) within large-scale systems exists due to baroclinicity and recently has been pointed out by Martineau et al. (2020) as an essential mechanism in the North Atlantic for large-scale system development.

Therefore, we hypothesize that the tilt within large-scale structures plays a role when calculating climate oscillations, which rely on one parameter at one specific atmospheric level (typically either at 500 hPa or at the surface). We thus suggest that composites of atmospheric and glaciological variables are intrinsically dependent on the climate oscillation in use.

LN76: It would be good to emphasize around this point in the introduction the explicit goal and primary research questions of the study. These would help build upon some previously mentioned hypotheses (e.g., LN 74-75) by adding more structure and thus guidance for the reader toward analyses that lie ahead.

Thanks, we changed this as suggested. We initially had decided to state each of our hypotheses next to the respective literature review paragraph (in the previous version: 1st hypothesis in LN 43 and 2nd hypothesis in LN 74) instead of a paragraph explicitly dedicated to our research goals. We will follow the Referee's suggestion to emphasize at the beginning of the last introductory paragraph our research aims.

Here, we explore a cluster method, called NAG, that links the role of the NAO with the prevailing mid-tropospheric circulation pattern over Greenland (GBI), along with the atmospheric water vapor over the GrIS. Since the NAG estimates the "influence" of large-scale systems over the North Atlantic on GrIS SEB components, we regionally investigate the climatology of atmospheric variables contributing to SEB for contrasting atmospheric circulation clusters. Finally, we examine changes of SEB components within clusters by comparing recent decades (1991-2020) to a historical period (1959-1990), with a special focus on the summer ablation zone

Section 2 describes the data analyzed, explains the clustering method, and justifies 1991 as the 62-year breakpoint. The section Results and Discussion is broken into three subsections. In Section 3.1 we present the inter-annual variability of the newly cluster-derived classification (NAG) and compare it with NAO and GBI alone; in Section 3.2 we describe the inter-seasonal and regional variability of the NAG; in Section 3.3 we study spatio-temporal anomalies within the same NAG phase, and finally, we concentrate our discussion on regional changes in the summer ablation zone.

LN 117: List the flux terms units, W/m^2?

Thanks, we included the flux terms units as suggested.

LN126-127: I think you could move this sentence (beginning with "Using the 62 years…") to L131 and explicitly list the sub-periods that were the result of equally dividing the total years in the dataset.

Thanks, we took into consideration the Referee's input! Since Referee #3 recommended to make this particular section clearer in the manuscript, we reformulated the paragraph. This improved paragraph contains an explanation of trends in sub-periods, a definition of the trend ratio, an example of one period ratio and most importantly we explicitly state the reason for the breakpoint detection.

The literature agrees that the pronounced Greenland summer mass loss started in the 1990s (e.g., Mouginot et al. 2019; Hanna et al. 2021; Shepherd et al. 2020). However, the onset of a clear negative trend varies depending on the time period of each study and on the dataset used. In order to determine the breakpoint of the marked summer surface mass loss in RACMO2, we divided the GrIS into its main seven drainage basins (see Fig. S1) and run 612 regional trends for periods with different lengths. For the 62 years of data, the length of sub-periods ranges from 15 (30-year period) to 32 years (62-year period). This will allow the investigation of atmospheric and glaciological conditions prior and post a potential breakpoint. The breakpoint was determined by assessing the most regionally frequent and the largest absolute trend ratios. One trend ratio (RT) is based upon two slopes from equally-sized sub-periods that are split in a common central year. RT is defined as the absolute value of the division between the slope after the central year (s2) and the slope before the central year (s1). For instance (central panel in Fig. 1), s1 between 1977 and 1995 and s2 between 1995 and 2013, whose central year is 1995 gives RT > 1. This means that s2 is more pronounced than s1.

The non-parametric Mann-Kendall (M-K) trend test is used to assess trend monotonicity and significance on summer surface ablation rates (c.f. Section 2.2). The slope corresponds to the Theil-Sen (T-S) estimator. The T-S estimator is a robust regression method that does not require the data to be normally distributed and is hence less vulnerable to outliers than conventional methods. One specific period is considered significant only when the confidence level from the M-K test is higher than (or equal to) 90% in both sub-periods. Trends in periods exhibiting confidence levels lower than 90% may still be identical to150 those exhibiting great significance levels but given their high variability they were not considered. The resulting combination of increasing (i) or decreasing (d) sub-period slopes is shown by color-coded cells (Fig. S2), whereas RTs are displayed in Figure 1.

As we analyzed 612 trends for equally-sized sub-periods, we consider it unfeasible to explicitly list them. Nevertheless, we show to the Referee in the figure below the distribution of sub-periods used.

[Figure]

Figure R1-Distribution of sub-periods used for the trend analysis in Section 2.3

L155: Both the NAO and GBI indices should be defined here (e.g., domains, methods, papers defining such, etc). Moreover, both atmospheric indices are derived at z500, did you look at the surface NAO (i.e., Hurrell PCA or weather station-based NAO)? In this context, I recommend in the paper that you address why only z500 indices are used or why NAO from SLP data is not used. This discussion would seem appropriate since you are exploring through cluster analysis how co-varying characteristics of these atmospheric patterns (along with IWV) may impact GrIS surface conditions.

Thanks, this indeed is a crucial remark to highlight in the methods, as the NAO can be calculated with different methods, different datasets at different pressure levels. We followed up on this by statistically comparing two datasets using PCA. You can find a better description of the products used in Section 2.4 and the NAG sensitivity as dependent on the NAO product in the Supplementary Material.

LN167-168: To clarify the sentence, I recommend substituting "predominant" with "prevailing" then remove "prevailing" in LN168.

Thanks, we changed this as suggested.

LN172: The clustering approach could use more description. Why did you pick 3 clusters? Did you select these based on subjective or objective criteria? Are the results sensitive to the number of clusters selected and analyzed?

The selection of 3 clusters (negative, neutral, and positive) is based on the number of climate oscillation phases commonly used in many studies (e.g. Croci-Maspoli et al. (2007), Gimeno, et al. (2002), Wu and Zhang (2015)). We believe that the inclusion of more clusters is not useful, as they become abstract and objectively challenging to physically interpret them. As shown above, the results are sensitive to the selected variables (NAO, GBI, GrIS IWV) in the cluster, as well as the number of clusters set. We made all these points clearer in the revised version as follows:

We design a classification called NAG, that estimates the "influence" of large-scale systems over the North Atlantic in Greenland, and is computed by applying k-means clustering to NAO, GBI, GrIS IWV. According to within-cluster sum of squares, a measure of variability within each cluster, the optimal number of clusters for our data is not larger than 3. Also, as climate oscillations are commonly identified as positive, neutral and negative phases, 3 clusters/classifications (+NAG, 0NAG, –NAG) were defined in advance. The 3 seasonal variables considered (NAO, GBI, GrIS IWV) are represented by 62 points/years in a 3-dimensional space. As an initial condition, 3 random points are selected in space to serve as the center of each cluster. The 3-dimensional Euclidian distances between the 62 points and the center of the 3 random clusters are calculated. Points are classified individually based on their distance to the center of the closest cluster. The center of the 3 clusters shifts iteratively by the mean distances of all points within its own cluster. The best possible grouping is achieved by selecting the minimum calculated sum of squares of the distances between grouped points and the mean center of each group. The k-means clustering method is then repeated seasonally. The resulting clustering classification (Fig. S3) is sensible on the choice of the time period, number of clusters defined and variables. A sensitivity analysis of the clustering and percentile classification using NAO (van den Dool et al., 2000) or NAO (Hurrell et al., 2003) and GBI is addressed in the Supplementary Material.

LN179: Be more specific on what data is shown in the Figure S3 scatterplots. This is very vague as currently written.

Thanks, we improved Figure S3 caption to:

Seasonal NAG phases based on the k-means clustering: positive (red cluster); neutral (black cluster); and negative (blue cluster) phase. NAO and GBI phases are categorized based on the respective seasonal 25th (blue line) and 75th (red line) percentile. The Spearman correlation coefficient (rS) is shown for clusters with significance higher than 90%. The third-dimension corresponding to GrIS IWV is omitted. The dark circle indicates years whose seasonal GrIS IWV is greater than the 95th percentile.

LN183: "The positive phase of NAG is connected…" Is this "connection" illustrated somewhere either graphically or statistically? This would be helpful to show the reader to see what +NAG entails.

We show the NAG and its relations in Figure 2, S3 and S4:

Figure 2 illustrates the development of NAO and GBI over time. Figure S3 shows the relationship between NAO and GBI and the respective NAG clusters/phases. Figure S4 characterizes the anomalies of geopotential height at 500 hPa with respect to the climatology dependent on season and NAG phase.

We completed the sentence with: The newly derived cluster classification (NAG, see Section 2.4) is based on the resulting clustering (Fig. S3) and the involved large-scale circulation (Fig. S4), the positive phase of NAG is connected to an anomalously high geopotential height at 500 hPa level (GBI > 0) as well as high IWV, and to the anomalously negative pressure difference between the semi-persistent Azores high and the semi-persistent Icelandic low (NAO < 0).

Figure 2 caption: The last sentence is unclear. I suggest mentioning that data from 1991 onward is found to the right of the gray vertical line.

We expand this and add:

Thanks! We improved the figure caption to:

Time-series of seasonal GBI, signal inverted NAO (iNAO) and NAG classification. GBI and iNAO phases are color-coded and NAG is coded by symbols shown positive (+); neutral (0); and negative (−) phase. The negative (positive) GBI and iNAO phase based on the 25th (75th) percentile are illustrated as diamonds. Seasonally accumulated surface mass balance (SMB) for absolute quantities larger than 200 Gt season−1 (winter DJF and summer JJA) is sized accordingly. A negative SMB is marked by a dark circle around the bubble. For reference, 1991 is highlighted as a gray vertical line to illustrate GBI, NAO and NAG phases.

L192: Why show the 925 hPa height anomaly rather than SLP, a field typically used in defining surface pressure characteristics of the NAO.

For practical reasons, model pressure levels were preferred, but for clarity we replaced 925 hPa by surface pressure in the revised version.

L193: "vertical tilting structure" meaning what? Please clarify.

We expand this definition stating:

Despite the typical life cycle of the NAO phase lasting about two weeks (Feldstein, 2003), the geopotential and temperature vertical tilting under strong baroclinicity described by Martineau et al. (2020) remains within seasonal composites (Fig. S4).

L199: Do you mean "winter" instead of "spring" is when the equator-to-pole air temperature contrast is maximized?

On average, the strongest equator-to-pole temperature gradient coincides with the end of winter. Depending on the year, the maximum equator-to-pole air temperature contrast takes place between February and March, and therefore coincides either in the meteorological late winter or early spring. In order to avoid misinterpretations, we drop the clause ("when the equator-to-pole temperature is the strongest") out of the sentence.

Figure 3: Should the colorbar label r_s(Seasonal ablation…) be r^2 as it is in the caption?

While the figure colors are representative of the Spearman correlation ($r_S$) and vary from -1 to 1, the noted number ($r_S^2$) indicates the proportion of shared variance among ranked variables and ranges between 0 and 1. We reformulated the second sentence of the figure caption to: The Spearman correlation coefficient ($r_S$) is color-coded and the determination coefficient ($r_S^2$) is displayed.

Figure 4: "The percentage of each NAG phase used..." can you please clarify what this means? As I interpret it, it sounds like some +/-NAG days were composited and some were not without explanation as to why since f0 and f-percentages do not sum to 100% (as in Fig 5).

Thanks for pointing this out. This was indeed misleading. The relative frequency of seasonal NAG (0NAG, +NAG, -NAG) sums to 100%. However, as the 0NAG is the reference for +NAG and -NAO, we indicated its frequency at both instances. We improve the mentioned caption sentence to: The relative seasonal frequency (f in %) of each NAG phase (+NAG, 0NAG and −NAG) used to produce composites is indicated as subtitle in a).

LN253: Remove "configuration."

Thanks, we removed this as suggested.

Also, since these surface temperature and radiative fields (i.e., Fig 5) increase regardless of atmospheric pattern, does that suggest that warming climate is the main culprit in driving these fluxes that impact SMB? What link is being made with adjacent marginal seas; they respond similarly to these fields as the GrIS?

Our results suggest warming climate and its spatial magnitude over the GrIS depending on the NAG phase. As we supposedly split internal climate variability into clusters, we point anthropogenic effects as the culprit of changes over land and overseas in the Conclusions. However, a direct attribution to individual factors of such a complex system is indeed very challenging and beyond our study scope.

Temporal changes over the adjacent seas depending on season and NAG phase are described and compared to GrIS along the Results and Discussion (e.g., LN261, LN279 and LN305). However, we acknowledge that this point was missing in the Conclusions, and therefore, we added it in a revised version (e.g., LN415).

LN 270-271: In comparing 1991-2020 against the reference period, do you mean "increased surface-based inversions…"

We simply adopted the hypothesis from other studies (e.g., Niwano et al. 2019) stating that surface-based inversions in combination with subsidence are the physical mechanism responsible of the elevated values of water vapor near the surface. To avoid confusion, we reformulate the sentence to: As a result of surface inversions favored by subsidence in association with surface melt, southern regions comprise more q2m for the period 1991-2020 than during the reference period (Fig. 5c).

LN272-273: This sentence is confusing; the IWV increase over the northern GrIS is not related to local cloud water content? Please clarify. Is this shown in the analysis and if so, then where?

The IWV increase is not only related to the local cloud water content changes, but also a result of the increasing near-surface water vapor. We do not attribute the increase in SWnet to cloud water content changes in the Northern regions because the SWin and cloud water content did not significantly change in comparison to the reference period (Fig. S12). We improve the sentence to: Despite the SW↓ (Fig. S12b) and the cloud water content little changed over the northern regions under +NAG, SWnet (Fig. S14) significantly increased mainly due to lower surface albedo. Noël et al. (2019) attributed the recent decrease in surface albedo over the northern regions to rising atmospheric temperatures and increased cloudiness. The LWP increase, which also contributes to the increase in LW↓ (Fig. 5b), is particularly pronounced in the Northwest regardless of the NAG phase.

LN293: "high summer GBI values…" –where is this analysis shown?

In order to reduce the number of supplementary figures, we removed seasonal GBI and NAO time-series. However, this statement is supported in Figure 2 as seen by the red shading enhancement and frequency of positive GBI greater than the third quartile (marked by diamonds). In addition, we also added a few studies entirely focused on the recent extreme GBI (e.g., Barrett et al. 2020; Hanna et al. 2016; Hanna et al. 2018)

LN 297: "crucial role of NAO advecting heat and moisture…" through storms/the storm track migrating poleward toward Greenland?

Thanks! We completed the sentence taking into consideration the reviewer's remark.

LN311-313: This sentence is a bit hard to follow. I recommend splitting it into two sentences.

Thanks! We follow the Referee's advice and changed this paragraph to: As the temperature of the melting snow/ice surface is physically limited to 273.15 K, the ΔT is essentially driven by the air temperature increase. The melting snow/ice surface, in conjunction with steep slopes, promotes downslope winds. In addition, contributions to the marked wind speed strengthening in the ablation zone arises from the migration of the snowline to higher elevations of the ice-sheet (Ryan et al., 2019), which in turn enhances the surface pressure gradient and adds momentum to the flow.

LN315: To clarify, is the suggestion that the summer wind speed increase over northern GrIS is due to the near-complete summer melt of Baffin (particularly) ice cover, a typical feature of its annual cycle? Atmospheric circulation patterns can accelerate the melt, but their intensity and orientation could also presumably affect the onset of such summer wind increases.

We agree with the Referee: atmospheric circulation patterns can influence the onset of the regional sea ice melt. We will add the Referee's thought as follows: Particularly in the northern regions, one factor that contributes to the increase in summer wind speed is the decrease in ice in the neighboring seas. The change in wind speed can be related to emerging open water feedback as it occurs irrespective of the prevailing atmospheric circulation pattern. The increase in wind speed favors polynya formation that generates low surface pressure over the open waters and hence enhances the regional surface pressure gradient. A thermal circulation is identified over the North and Northeast Greenland, where margins are almost permanently ice-covered during the reference period. In the Northwest ablation zone, the wind speed has not increased significantly potentially related to the seasonal Baffin Bay ice-free.

LN 322-324: It would be a good idea to direct the reader to this figure or analysis within the paper.

Thanks! We followed the advice and add the figure (Fig. 6) in the main paper in the revised version.

LN382: Change "has" to "have"
Thanks, we changed this as suggested.
**Supplemental Material:**
Figure S8: Label seasons at the top of the graphic as with Figure S7, etc.
Thanks, we labeled this as suggested.
Figures S12-14: I am confused what these graphics actually show and what the units on each concentric circle represent. Please clarify.
Thanks, we expand their description in their caption stating:
Fig. S12. Changes in atmospheric variables contributing to SEB components between 1959-1990 and 1991-2020 in the summer ablation zone for each NAG phase (color-coded). Variables in the panel a) exhibit the variable name and the respective unit. All variables in the panel b) are fluxes in W m$^{-2}$. The spatial relative frequency of ablation is shown at the center. Negative changes are limited by the gray area. Hollow circles indicate significant mean differences based on the Wilcoxon rank-sum statistic test for unpaired sets with a confidence higher than 90%.
These charts come to support the analysis in Figure 6. The radar chart aspect is an alternative to the bar chart in Figure 6.

**References:**

- Barrett, B. S., Henderson, G. R., McDonnell, E., Henry, M., and Mote, T.: Extreme Greenland blocking and high-latitude moisture transport, Atmospheric Science Letters, 21, e1002, https://doi.org/10.1002/asl.1002, 2020.
- Croci-Maspoli, Mischa, Cornelia Schwierz, and Huw C. Davies. "Atmospheric blocking: Space-time links to the NAO and PNA." Climate Dynamics 29.7 (2007): 713-725.
- Gimeno, Luis, et al. "Identification of empirical relationships between indices of ENSO and NAO and agricultural yields in Spain." Climate research 21.2 (2002): 165-172.
- Hanna, Edward, et al. "Greenland blocking index daily series 1851–2015: Analysis of changes in extremes and links with North Atlantic and UK climate variability and change." International Journal of Climatology 38.9 (2018): 3546-3564.
- Hurrell, James W., et al. "An overview of the North Atlantic oscillation." Geophysical Monograph-American Geophysical Union 134 (2003): 1-36.
- Niwano, Masashi, Akihiro Hashimoto, and Teruo Aoki. "Cloud-driven modulations of Greenland ice sheet surface melt." Scientific reports 9.1 (2019): 1-8.
- Noël, Brice, et al. "Rapid ablation zone expansion amplifies north Greenland mass loss." Science Advances 5.9 (2019): eaaw0123.
- Ryan, J. C., et al. "Greenland Ice Sheet surface melt amplified by snowline migration and bare ice exposure." Science Advances 5.3 (2019): eaav3738. https://doi.org/10.1126/sciadv.aav3738
- Van den Dool, H. M., S. Saha, and AAke Johansson. "Empirical orthogonal teleconnections." Journal of Climate 13.8 (2000): 1421-1435.
- Wu, Zhiwei, and Peng Zhang. "Interdecadal variability of the mega-ENSO–NAO synchronization in winter." Climate dynamics 45.3 (2015): 1117-1128.
- Wachowicz, L. J., Preece, J. R., Mote, T. L., Barrett, B. S., and Henderson, G. R.: Historical Trends of Seasonal Greenland Blocking Under Different Blocking Metrics, Int J Climatol, 41, E3263–E3278, https://doi.org/10.1002/joc.6923, 2020.

**Comment on tc-2021-388**

Anonymous Referee #2
Referee comment on "The impact of climate oscillations on the surface energy budget over the Greenland Ice Sheet in a changing climate" by Tiago Silva et al., The Cryosphere Discuss., https://doi.org/10.5194/tc-2021-388-RC2, 2022

*This is an interesting novel study of the effects of the NAO and GBI and their combined influence on the Greenland Ice Sheet surface energy balance. The paper is fairly insightful and is reasonably well presented overall, although it would be useful to add an explanation of exactly how the NAG (influence of the North Atlantic over Greenland) time series was derived. It should also be clarified somewhere whether the reported correlation coefficients are based on detrended datasets. The analysis is based on an interesting and worthwhile hypothesis that the tilt within North Atlantic jet stream structures may have differing spatial (and temporal) effects on the near-surface impacts of jet-*

*stream changes, and is best quantified by combining NAO and GBI rather than taking one of these measures in isolation.*

We thank Referee #2 for the constructive review and the overall positive assessment of our study. We acknowledge that our methodological description of the derivation of NAG was apparently not clear enough in the initial submission. We believe that we overcame this by reformulating the entire paragraph dedicated to the NAG explanation in Section 2.4.

**Specific comments**
Line 92 (P4): why not use the most recent (and best) ERA5 ECMWF reanalysis (which is available back to 1950) to force the RACMO for the whole time period?
Thanks, we completed LN92 with:
We used the latest version of RACMO2.3p2, which is forced by a combination of ERA reanalyses: ERA-40 (1958-1978), ERA-Interim (1979-1989) and ERA5 (1990-2020).
Line 97 (P4) "based on the lowest 5% albedo values between 2000 and 2015" – how many values/how frequent?
Thanks, we expanded the mentioned sentence to: Bare ice albedo is estimated as the $5^{th}$ percentile of the recorded albedo in each year by the 16-day MODIS product (MCD43A3) over the period 2000-2015. The resulting annual maps of MODIS-derived bare ice albedo are then averaged over the period 2000-2015 (Noël et al. 2018, 2019). The analysis of how many values/how frequent goes beyond our study scope.
L162 (P7) etc. – are the reported correlation coefficients based on de-trended data?
The reported correlations are not detrended, as none of the used variables are deterministic. Nevertheless, if we had assumed that the variables analyzed along L162 were linearly related, the correlation between detrended variables would be weaker but with the same sign. In order to overcome the linear relationship assumption, we do not use Pearson correlation coefficient but a non-parametric technique, namely Spearman correlation coefficient.
L183 (P8): how exactly is the NAG time series calculated?
As explained in the last paragraph in Section 2.4, the NAG is calculated by k-means clustering using NAO, GBI and GrIS IWV. We add: (see Section 2.4) to the text, now, in L211. Additionally, we also improved clustering description, as recommended by the Referee #1 and Referee #3 (LN195).
L187 (P8) "the influences exerted by NAO and GBI may differ" – this is an interesting result.
Thanks for this positive assessment! We improved the discussion of the differences of the regional changes with respect to the reference period by using only NAO or only GBI in most atmospheric variables (Fig. S10 and S11).
L190 (P8) "the $95_{th}$ percentile of IWV is mainly connected to positive NAG phases in summer and winter": Fig. S3 suggests (dark circles) that high IWV is mainly associated with the neutral (grey) cluster then.
We improved Figure S3 in order to avoid confusion as pointed out by the Referee.
L198 (P8) "in winter +NAG frequently contributes the most to surface accumulation". What is the reason for this? If +NAG means more Greenland Blocking, this should be associated with fewer storms in south-east Greenland.
NAG is the combination of GBI, NAO and GrIS IWV. One seasonal +NAG phase does not necessarily indicate only one specific blocking situation centered over Greenland, but rather the influence of a series of high-pressure systems nearby Greenland interrupted by episodic cyclonic activity (e.g., Bjørk et al., 2018; Hanna et al. 2018, Woollings et al. 2008). We mentioned in the Introduction how wave breaks in the North Atlantic contribute to surface mass gains in the western part of Greenland (LN 55).
Fig. 3 (and elsewhere): are the reported correlation coefficients based on de-trended data?
Since we segregate the data by season and NAG phase, each sample is typically composed by assorted years. This means that the following value is not dependent on the previous value.
If we had assumed that the relationship among variables for the 62 years in the analysis were linear and deterministic, the detrended version would not change the results, but only attenuates the correlation coefficients. This attenuation makes a few weak detrended relationships weaker and not significant without changing the correlation coefficient ranks among significant correlations.
If we had assumed that the relationship among variables in analysis were non-linear and stochastic within the segregated data, we would be able to detrend the data by differencing. The resulting correlations would be strengthened in comparison with the raw data. However, both detrended methods strongly rely on subjective decisions which is why we present raw non-parametric correlations.
L238 (P12) "-NAG in winter promotes more IWP at the Northeast" – this seems unclear from Fig. 4(a).
We agree that this information cannot exclusively be seen in Fig 4a and in addition Figure S8d has to be taken into account. We add the reference to this figure in the revised version. Given the negative winter temperatures, the increase in cloud content over the Northern regions is entirely attributed to increases in IWP.

L240 (P12) "The RH2m…" – where is this shown? Should this refer to the q2m plots?

RH2m is not shown, and we now acknowledged that RH2m does not add much to the discussion. We removed these unnecessary details in the revised version.

**References:**

- Bjørk, A., Aagaard, S., Lütt, A., Khan, S., Box, J., Kjeldsen, K., Larsen, N., Korsgaard, N., Cappelen, J., Colgan, W., et al.: Changes in Greenland's peripheral glaciers linked to the North Atlantic Oscillation, Nature Climate Change, 8, 48–52, https://doi.org/10.1038/s41558-017-0029-1, 2018.
- Hanna, Edward, et al. "Greenland blocking index daily series 1851–2015: Analysis of changes in extremes and links with North Atlantic and UK climate variability and change." International Journal of Climatology 38.9 (2018): 3546-3564.
- Noël, Brice, et al. "Rapid ablation zone expansion amplifies north Greenland mass loss." Science Advances 5.9 (2019): eaaw0123.
- Woollings, Tim, et al. "A new Rossby wave–breaking interpretation of the North Atlantic Oscillation." Journal of the Atmospheric Sciences 65.2 (2008): 609-626.

**Comment on tc-2021-388**

Anonymous Referee #3
Referee comment on "The impact of climate oscillations on the surface energy budget over the Greenland Ice Sheet in a changing climate" by Tiago Silva et al., The Cryosphere Discuss., https://doi.org/10.5194/tc-2021-388-RC3, 2022

**General comments**

*In this study, the authors use a cluster analysis of NAO, GBI, and column water vapor to derive a "North Atlantic influence on Greenland" (NAG) index. RACMO2 output is then used to investigate atmospheric and cryospheric conditions across different NAG phases and their changes across a 1991 break point in summer surface mass loss. Results describe a large array of seasonal anomalies in atmospheric conditions and surface energy balance components across the NAG phases for each season during the pre- and post-1991 periods.*
*I found this paper difficult to follow due to the large number of figures and sub-panels within figures and the organization of the paper, as it lacks a clear statement of the research questions or summary of what important new information was learned in this study. I also agree with the editor that there is insufficient originality, at least with how the results are presented in current form. However, there do not appear to be any technical flaws in the methods employed, and I do think there is potential for some of the results to form a nice study if they are better organized. I encourage the authors to think about what they consider to be the most important and novel findings contained within their many analyses and distill these findings into a focused message for readers to take from the paper. As an example, the contrast between moistening in northern Greenland and drying / clearing in southern Greenland under +NAG conditions is an interesting finding. In addition to the specific comments and technical corrections below, I would recommend that the authors simplify the figures, and restructure the discussion so that a large part of the findings in the main paper are not describing figures found in the supplement.*

We thank Referee #3 for the challenging remarks and appreciate the overall potential they see in our study. We acknowledged that there was indeed an obviously too unclear aim of the study which was similarly pointed out by Referee #1 and we implemented this very thoroughly in the revised version. We also took up the Referee's recommendation and will strengthen the discussion in the revised version. This will significantly improve the reading of the paper. For now, we would point to the amendments made in revised version point-by-point.

**Specific comments**

Did the authors examine trends in the frequency of NAG phases, or did they only look at changes in atmospheric conditions over time during each NAG phase?

[Figure]

Figure R2 - 10-year running of NAG fractions

We made a brief trend analysis in the frequency of the NAG phases. We collected decadal-moving fractions as dependent on the NAG phase (Figure R2). However, we solely focused on spatio-temporal changes of the atmospheric conditions within NAG phases overtime. The NAG development over time is found in manuscript Figure 2.

L18–45: The opening paragraph of the Introduction is quite long and does not provide a compelling introduction to the research topic that the authors investigate. I think it would make more sense to first introduce the problem of Greenland surface melt and its atmospheric drivers (the second paragraph), before moving on to the indices that are used to help quantify these atmospheric drivers (first paragraph).

We acknowledge the Referee's comment, which was also mentioned by Referee #1. We reformulated the Introduction accordingly by moving the first paragraph and breaking it into two parts in the revised version.

L21: Liu and Barnes (2015) is a good reference on the relationship between Rossby wave breaking and poleward moisture transport in the vicinity of Greenland.

Thanks for pointing us towards this reference which we implemented in the introduction of the revised version (e.g., LN66).

L29: I'm not sure it's correct to say that the NAO phase "explains most of the heat and moisture transported poleward". It's more accurate to say that the NAO phase affects the location and magnitude of poleward heat and moisture transport, and provide a reference on this.

Indeed, the formulation was misleading, and we adopt the formulation suggested by the referee. At the same time, we add the following references (e.g., Bjørk et al., 2018, Papritz et al, 2020).

L32: GBI simply quantifies the mean 500 hPa geopotential height over a Greenland centered domain, as the authors state in the previous sentence. It does not directly quantify the strength and moisture transported over the Greenland domain although it is correlated with these quantities (see the Barrett et al. 2020 paper the authors already cite). See Wachowicz et al. 2020 for a more nuanced discussion of the GBI and comparison with other blocking metrics.

To our understanding, GBI provides insight on blocking-like conditions in the vicinity of Greenland. We agree that "quantify the strength" was not the best wording. Nevertheless, positive GBI conditions can regionally block heat and moisture transport towards the interior of the GrIS. We reformulate the sentence as: Its index denotes the predominant atmospheric circulation pattern in the vicinity of Greenland, and it regionally controls the heat and moisture transported towards the interior of the GrIS.

L120 and Figs. 5, S9–S11: It is not clear how the method of dividing the adjacent seas into four areas is actually used to assess potential sources of moisture. I am having trouble understanding what the numbers in the corners of Figs. 5 and S9–S11 (the "differences in composites between adjacent seas") represent.

We were interested to know if changes in atmospheric variables over land were similar to the changes over the adjacent seas, as moisture and heat are advected from there towards the ice sheet. In order to simplify the visualization, instead of showing the spatial anomalies over the adjacent seas we preferred to show the temporal change in one value, which

is displayed in the corners of each subpanel. We acknowledge the complexity of the figure, but at the same time we value its contribution to the discussion. We moved the 0NAG to supplementary material and improved the caption description to: Seasonal and spatial anomalies for (a) integrated water vapor, (b) incoming longwave radiation reaching the surface, (c) near-surface specific humidity, and (d) skin temperature from RACMO2 between 1991-2020 and 1959-1990 as dependent on the NAG phase. The percentage (f) of the NAG phase in each period is indicated for each season. For reference, Summit and South Dome are marked as big and small triangles, respectively. Stippled regions indicate areas with a confidence level greater than 90% (based on the Wilcoxon rank-sum statistic test for unpaired sets). Temporal anomalies between composites over the adjacent seas are also shown as colored numbers (Baffin Bay: upper left; Greenland Sea: upper right; Irminger Sea (lower right) and Labrador Sea (lower left). Temporal anomalies equal to null are omitted. See Figure S1 to discern the extension overseas and Figure S9 to examine seasonal and spatial anomalies under 0NAG.

L123: It should be stated explicitly at the beginning of section 2.3 that the reason for the break point detection is to form the basis for subsequent analyses of atmospheric and glaciological conditions before and after the break point. As it stands now, this section reads like it is reporting research findings, rather than describing a method that will be used to produce the results of the study.

We fully agree with the comment. Also, taking into consideration the comment from Referee #1, we reformulated the beginning of Section 2.3 to:

The literature agrees that the pronounced Greenland summer mass loss started in the 1990s (e.g., Mouginot et al. 2019; Hanna et al. 2021; Shepherd et al. 2020). However, the onset of a clear negative trend varies depending on the time period of each study and on the dataset used. In order to determine the breakpoint of the marked summer surface mass loss in RACMO2, we divided the GrIS into its main seven drainage basins (see Fig. S1) and run 612 regional trends for periods with different lengths. For the 62 years of data, the length of sub-periods ranges from 15 (30-year period) to 32 years (62-year period). This will allow the investigation of atmospheric and glaciological conditions prior and post a potential breakpoint. The breakpoint was determined by assessing the most regionally frequent and the largest absolute trend ratios. One trend ratio (RT) is based upon two slopes from equally-sized sub-periods that are split in a common central year. RT is defined as the absolute value of the division between the slope after the central year (s2) and the slope before the central year (s1). For instance (central panel in Fig. 1), s1 between 1977 and 1995 and s2 between 1995 and 2013, whose central year is 1995 gives RT > 1. This means that s2 is more pronounced than s1.

The non-parametric Mann-Kendall (M-K) trend test is used to assess trend monotonicity and significance on summer surface ablation rates (c.f. Section 2.2). The slope corresponds to the Theil-Sen (T-S) estimator. The T-S estimator is a robust regression method that does not require the data to be normally distributed and is hence less vulnerable to outliers than conventional methods. One specific period is considered significant only when the confidence level from the M-K test is higher than (or equal to) 90% in both sub-periods. Trends in periods exhibiting confidence levels lower than 90% may still be identical to150 those exhibiting great significance levels but given their high variability they were not considered. The resulting combination of increasing (i) or decreasing (d) sub-period slopes is shown by color-coded cells (Fig. S2), whereas RTs are displayed in Figure 1.

L172–180: State up front that you are using a k-means clustering method (rather than first describing the method and naming it as k-means clustering at the end of the description).

Thanks, we named the clustering method before its description.

L181: I don't think the "influence of the North Atlantic over Greenland" is an accurate description of what the NAG index produced by the cluster classification provides. Maybe describe as the "influence of regional climate" on Greenland instead. (The AMV index, which specifically quantifies oceanic conditions, is discussed in the Introduction and in L159 in the Data and Methods, but doesn't appear to be included as an input to the NAG index.)

We name NAG as such given the importance of the North Atlantic Oscillation in setting the storm track and atmosphere-ocean interactions feeding large-scale systems along the North Atlantic. As mentioned in the Introduction, the AMV index was negative until the early 1990's and it has been positive since. Given the strong relationship in summer between AMV and GBI (and GrIS IWV), the inclusion of AMV alongside NAO, GBI and GrIS IWV in the 62 years of cluster analysis does not contribute with new information and it does not change the summer classification. However, in the remaining seasons (e.g. spring), positively high AMV in recent years adds noise to the classification by making GBI<0 and NAO>0 as +NAG, which is contradicting with the rest of the cluster.

L226, 232–236, 311–315, and 368–370: The authors should consider that the stronger wind speeds during the +NAG phase are not strictly katabatic but are enhanced by the interaction of a strengthened synoptic-scale pressure gradient with the Greenland ice sheet's orography. I would suspect this is especially true for the winter cases where the authors find that increased wind speeds and SHF occur during +NAG. Previous studies have described this synoptically-driven wind enhancement as the Greenland "barrier jet" or "plateau jet" – see e.g. Meesters 1994, van den Broeke and Gallée 1996, Moore et al. 2013, Mattingly et al. 2020.

Thanks, we took into consideration the Referee's input! We believe that the discussion regarding the coupling of the katabatic winds with upper air winds definitely improved the manuscript discussion(e.g., LN262).

L255, Figs. 4–6: I assume all the results in Figs. 4–6 (e.g. the increasing trend in TCWV in northern Greenland described in L255) are produced from RACMO2 data? If so this should be explicitly stated in the figure captions and the text.

We added to the figure caption that these results are based on the RACMO2.3p2 output, and convenient figure reference along the text.

L282–284: This statement about the seasonal preconditioning effect of skin temperature warming appears to contract the finding in L166–168 that there is no relevant time-lag response between seasonal GrIS surface mass fluxes and the predominant atmospheric circulation pattern prevailing in the preceding seasons.

The cross-correlation in L166–168 was applied to the overall GrIS SMB with isolated climate oscillations. In L282–284, we discuss that the skin temperature in winter during the period 1991-2020 is warmer than in 1959-1990 independently of the NAG phase. Hence, we do not attribute the impact of changing Tskin during individual subsequent seasons, which is why a direct comparison among the statements is not possible. We could add a clarification as "Note, that Fig. 5 compares general conditions among different periods and cannot be used in order to deduce subsequent season's cause and effect". If the Referee does not find this answer satisfactory, we kindly ask you to reformulate how can these statements be contradicting.

L314: How would decreasing ice in neighboring seas contribute to an increase in summer wind speed? Please explain in more detail.

Thanks, we took into consideration the Referee's suggestion by better explain the emerging open waters feedbacks, now in LN354.

The change in wind speed can be related to emerging open water feedback as it occurs irrespective of the prevailing atmospheric circulation pattern. The increase in wind speed favors polynya formation that generates low surface pressure over the open waters and hence enhances the regional surface pressure gradient. A thermal circulation is identified over the North and Northeast Greenland, where margins are almost permanently ice-covered during the reference period. In the Northwest ablation zone, the wind speed has not increased significantly potentially related to the seasonal Baffin Bay ice-free.

**Technical corrections**

L2: The word "fluxes" is not needed since "advection" already describes the horizontal flow of heat and moisture.

Thanks, we changed this as suggested.

L2: surface mass balance of what? (state definitively that it's the SMB of the Greenland Ice Sheet)

Thanks, we changed this as suggested.

L2: "pattern" --> "patterns"

Thanks, we changed this as suggested.

L14: "optical" --> "optically"

Thanks, we changed this as suggested.

L14–16: Run-on sentence. Consider splitting into two sentences.

We follow the advice and split up the sentence in two:

Increases in net shortwave radiation are mainly connected to high-pressure systems and their drivers are regionally different. In the southern part of Greenland, the atmosphere has gotten optically thinner due to the decrease in water vapor thus allowing more incoming shortwave radiation to reach the surface. However, we find evidence for southern regions where changes in net longwave radiation balance changes in net shortwave radiation, suggesting the turbulent fluxes control the recent SEB changes.

L15: "shortwave radiation flux" should be "shortwave radiation" or "shortwave radiative flux"

Thanks, we changed this as suggested.

L18: north of the *climatological location of* the jet stream

Thanks, we clarify this ambiguity by changing the sentence to:

The GrIS is most commonly found north of the jet stream

L63: "largest" --> "most intense"?

Thanks, we follow this advice and reword accordingly

L91: ERA5 is the most recent reanalysis product from ECMWF (it's not an "earlier product")

Thanks, we agree with the Referee and reword to: The ECMWF reanalyses products - ERA40 (Uppala et al., 2005) (1959-1978); ERA-I (1979-1989); and ERA5 (1990-2020) - are used to laterally force…

L211: The abbreviation "0NAG" is used repeatedly from this point forward without previously being defined in the text. It appears to be defined in the caption for Figure 4, but its meaning should be explicitly stated in the text at first use.

Thanks, we add its meaning to the beginning of Section 3.2: Spatial and inter-seasonal anomalies under contrasting NAG (+/−) phases with respect to the neutral phase (0NAG) are illustrated in Figure 4 (and Fig. S7) for IWV, incoming longwave radiation (LW↓), specific humidity at 2 m (q2m) and skin temperature (Tskin). Seasonal Tskin and the air temperature at 2 m(T2m) are highly and positively correlated (rS>0.9) in the ablation and accumulation zones for contrasting NAG phases.

L331: Delete the word "or" at the end of this line.

Thanks, we deleted "of" at the end of this line.

L335: Accumulation zone has been decreasing *in area*?

Yes, in area, we add this at the respective line in order to clarify.

L337: Insert the word "zone" after "accumulation"

Thanks, we changed this as suggested.

L366-367: "vertically distributed changes" --> "vertical distribution of changes"

Thanks, we changed this as suggested.

**References**

Liu, C. and Barnes, E. A.: Extreme moisture transport into the Arctic linked to Rossby wave breaking, J. Geophys. Res. Atmos., 120, 3774–3788, https://doi.org/10.1002/2014JD022796, 2015.

Mattingly, K. S., Mote, T. L., Fettweis, X., van As, D., Van Tricht, K., Lhermitte, S., Pettersen, C., and Fausto, R. S.: Strong Summer Atmospheric Trigger Greenland Ice Sheet Melt through Spatially Varying Surface Energy Balance and Cloud Regimes, J. Climate, 33, 6809–6832, https://doi.org/10.1175/JCLI-D-19-0835.1, 2020.

Meesters, A.: Dependence of the energy balance of the Greenland ice sheet on climate change: Influence of katabatic wind and tundra, Q.J Royal Met. Soc., 120, 491–517, https://doi.org/10.1002/qj.49712051702, 1994.

Moore, G. W. K., Renfrew, I. A., and Cassano, J. J.: Greenland plateau jets, Tellus A: Dynamic Meteorology and Oceanography, 65, 17468, https://doi.org/10.3402/tellusa.v65i0.17468, 2013.

van den Broeke, M. R. and Gallée, H.: Observation and simulation of barrier winds at the western margin of the Greenland ice sheet, Q.J Royal Met. Soc., 122, 1365–1383, https://doi.org/10.1002/qj.49712253407, 1996.

Wachowicz, L. J., Preece, J. R., Mote, T. L., Barrett, B. S., and Henderson, G. R.: Historical Trends of Seasonal Greenland Blocking Under Different Blocking Metrics, Int J Climatol, 41, E3263–E3278, https://doi.org/10.1002/joc.6923, 2020.

---

## Author Response (AR2)

We would like once more to thank the editor and the three anonymous referees for their constructive comments along the reviewing process that undoubtfully improved the manuscript. In the following, we will address the referees' comments point by point. We mark red the comments given by the referee, give our answers and comments in black and indicate how we addressed the amendments in the manuscript in green.

**Comment on tc-2021-388**
Anonymous Referee #1
Referee comment on "The impact of climate oscillations on the surface energy budget over the Greenland Ice Sheet in a changing climate" by Tiago Silva et al., The Cryosphere Discussions

*I have read the responses and tracked revisions in "The impact of climate oscillations on the surface energy budget over the Greenland Ice Sheet in a changing climate" by Silva et al. The authors have made an effort to address my concerns and improve the paper, especially by re-writing much of the abstract and introduction to clarify the intent of the study and its main outcomes. My remaining comments are aimed at attaining further clarity, especially toward describing the cluster methodology and results stemming from this tool.*

General comment:
The manner of describing and presenting the cluster method and developing the classification is still a bit confusing. I suggest in the abstract and elsewhere clearly stating that the k-means clustering method (applied to GBI, NAO, IWV, etc) was used to create the NAG index. For ease of interpretation I suggest referencing the cluster "method" or "classification" (these are synonymous terms), and describing the optimal cluster solution for this analysis as three clusters (which describe the spectrum of the NAG index, i.e., positive, neutral, and negative NAG). Referring to each of these three clusters as a "classification" is confusing (e.g., Lines 247, 250, etc).

Thanks for the general comment. We now refer to k-means clustering and optimal clustering solution in the abstract as:

By using k-means clustering, we name the combination of the Greenland Blocking Index and the North Atlantic Oscillation index with the vertically integrated water vapor as NAG. NAG captures the influence of atmospheric circulation patterns from the North Atlantic on Greenland with the optimal solution of three clusters (positive, neutral and negative phase).

In addition, we revisited all instances in the manuscript where NAG is mentioned in order to be more coherent with the naming of the clusters along the manuscript.

These lines (L) below reference the tracked version of the manuscript.

L22: Would "intensified" rather than "reinforced" be a better word choice here?
Thanks! The word "intensified" is indeed a better choice.
L113: "The vertical tilt of temperature and pressure within large-scale systems" may make this more clear than simply "geopotential."
Thanks, we changed this as suggested.
L118: "dependent on phase of the concurrent climate oscillation" or similar wording is recommended.
Thanks, we changed this as recommended.
L120-121: Again, for simplicity in describing your analytical approach I recommend a revision such as "To overcome this dependency on one atmospheric index, we use a cluster method to derive the NAG index, which groups phases of the NAO and GBI with…"
Thanks for the suggestion. The simplified sentence reads:
To overcome this dependency on one atmospheric index, we apply a cluster method to derive the NAG by using NAO, GBI and the atmospheric water vapor (IWV) over the GrIS. Therefore, NAG links the role of the NAO with the prevailing mid-tropospheric circulation pattern over Greenland (GBI), along with the IWV over the GrIS.
L172: "All terms are in Wm^-2, and represent…"

Thanks, we changed this as suggested.

L234-235: By using Spearman's approach, dataset normality is not assumed a priori. Unless certain variables are correlated with time, I would encourage omitting the sentence "In such a way, no trend in the data is assumed a priori" as the presence of trend is not assumed by using this method, but rather is tested for using the Mann-Kendall approach. These assumptions should be clarified and if detrending is conducted prior to calculating Spearman's correlations that should also be stated.

Thanks for the clarification, we fully agree with Referee and removed the sentence as suggested.

L279: Do you mean "500 hPa geopotential height and surface temperature (or pressure?) vertical tilting." – please clarify.

Thanks for the remark. The word "height" was missing in the sentence. Also, it is indeed misleading to mention temperature vertical tilting when only geopotential height at 500 hPa and surface pressure are examined. Surface pressure in Figure S4 comes by suggestion of Referee #1 in the previous revision round, but since the result is the same, we kept the statement with the "geopotential height tilt" for simplicity. The sentence reads now as:

Despite the typical life cycle of the NAO phase lasting about two weeks (Feldstein 2003), the geopotential height vertical tilting described by Martineau et al. (2020) remains within seasonal composites due to strong baroclinicity (Fig. S4).

L508: "not captured by isolated indices" may be more clear

Thanks, we changed this as suggested.

**Comment on tc-2021-388**

Anonymous Referee #3

Referee comment on "The impact of climate oscillations on the surface energy budget over the Greenland Ice Sheet in a changing climate" by Tiago Silva et al., The Cryosphere Discussions

Summary and remaining comments

*Thanks to the authors for their thorough responses to my comments and those of the other two reviewers. The paper has been improved considerably and I am satisfied with the authors' responses to most of my specific comments.*

*I feel the main aspect that needs further improvement before publication is the abstract, which in my opinion is still rather aimless and lacking clear statements of the main goals and findings of the study. If I am interpreting the results correctly, the authors have found that (1) the large-scale North Atlantic climate conditions have had a major impact on recent Greenland warming (with the North Atlantic influence more pronounced in certain regions and seasons); and (2) strong warming has occurred in some regions and seasons across all NAG phases, which the authors interpret as an indication that more localized influences (i.e. sea ice loss) have also contributed to Greenland warming. I think the abstract would be more effective if it were organized more clearly around these large-scale and local-scale contributions to Greenland mass loss and their interactions. The authors do a good job of describing potential interactions between the large-scale North Atlantic conditions and regional sea ice conditions in L47-54 of the introduction.*

*I have also listed several remaining technical corrections that I think the authors should make.*

Thanks for the remaining comments! We now organized the abstract in a way that large-scale (NAG) and local-scale (sea ice loss) influences on GrIS are separated.

Technical corrections

L10: The meaning of the phrase "for contrasting NAG phases" here is unclear. Does this mean the atmosphere has become warmer and moister across all NAG phases?

Thanks, the word "across" suits much better than the previous wording.

L20: "gotten" --> "become"

Thanks, we changed this as recommended.

Section 2.3: State up front (e.g. in L140) that the variable used to calculate trends in mass loss is the surface integrated ablation rate during summer (this is stated in the Fig. 1 caption and in L165, but it should be stated clearly when introducing the method in the text as well).

Thanks, we now included the variable stated in the Figure 1 in L140 and it reads:

In order to determine the breakpoint of the marked summer surface mass loss in RACMO2, we divided the GrIS into its main seven drainage basins (see Fig. S1) and regionally calculate 612 trends of the summer surface integrated ablation rate for periods with different lengths.

L176: "The used product" --> "This product"

Thanks, we changed this as suggested.

L177: based on *a* specific surface station

Thanks, we included the missing indefinite article.

L179: Restate here that GBI is derived from 500-hPa height over the region (60N-80N, 80W-20W)

Thanks, we changed this as suggested.

L209: If I am interpreting the supplement correctly, the distinction between the van den Dool and Hurrell NAO methods is that van den Dool uses 500 hPa height and Hurrell uses surface pressure? It would be helpful to state this difference between the two methods at this point in the paper text.

Thanks, although we pointed it out in the first sentence in Section 2.4, we stress this difference once more in the last sentence of the same section, and it reads:

A sensitivity analysis of the clustering and percentile classification using NAO (van den Loon et al. 2000) derived from 500 hPa geopotential height or NAO (Hurrel et al. 2003) derived from surface pressure and GBI is addressed in the Supplementary Material.

L213-216: I'm not sure of the purpose of the first paragraph in section 3.1. I think it could be removed.

Thanks, we partly agree with Referee #3. However, we would like to stick with most of the first sentence in Section 3.1, as often readers jump the Methods (+Supplementary Material) and dive immediately into the Results. In this case, they are informed about the large-scale resemblances among indices used in the study.

L269: due to *the* steep surface

Thanks, we changed this as suggested.

L429: shorthwave --> shortwave

Thanks, we changed this as suggested.

Responses to specific author comments

Thanks for the clarification on my comment on L120 about averaging variables over adjacent seas. It is now clear that the point of this analysis is not to quantify the magnitude of moisture sourced from these seas, but rather to assess whether NAG influences on atmospheric conditions in these seas are similar to the adjacent sectors of the ice sheet.

Thanks for the clarification on my comment on L282-284 about the seasonally lagged effect of winter skin temperature warming. I now understand the argument the authors are making, and I think the introduction of the statement "Nevertheless, the NAG phase in summer governs the overall surface mass loss" in L318-319 helps the reader understand that the authors are not explicitly connecting winter skin temperature warming in individual seasons to enhanced mass loss in the subsequent summer.